# BASIL: Scalable Bayesian Semi-supervised Clustering with Feature Selection and Adaptive Constraint Weighting

**Luwei Wang** [1]  **Dagmara Panas** [1]  **Ke Wang** [1]  **Bruce Guthrie** [2]  **Sohan Seth** [1]

## Abstract

Constrained clustering incorporates prior knowledge in the form of pairwise constraints to guide data partitioning. While effective, existing Bayesian approaches are often limited in scalability to large datasets and provide weak interpretability due to the lack of explicit feature relevance modeling. We propose BASIL, a scalable Bayesian semi-supervised clustering framework that leverages stochastic variational inference to jointly infer cluster assignments and feature importance weights. This joint formulation enables the identification of discriminative features consistent with the imposed constraints. To robustly handle noisy or inconsistent supervision, BASIL introduces an adaptive constraint-weighting mechanism that down-weights unreliable constraints. Experiments on synthetic and real-world benchmarks show BASIL attains competitive accuracy while reducing training time by over $96\%$ on large datasets, learns interpretable cluster-specific feature importance maps, and remains robust to up to $30\%$ noisy constraints under sufficient supervision. We further demonstrate applicability to large-scale health data, including medical imaging and electronic health records.

## 1. Introduction

Clustering partitions observations into groups such that samples within a cluster are more similar than those across clusters (Velmurugan & Santhanam, 2011). While traditional clustering operates unsupervised, in many scenarios prior knowledge about cluster structure is available. Semi-supervised clustering methods leverage both unlabelled data and limited supervision to discover outcome-driven struc-

tures (Bair, 2013). When the true number of clusters is unknown or only partial supervision is available, a common form of supervision involves pairwise constraints (Bilenko et al., 2004). These constraints encode prior knowledge in the form of must-link (two samples must belong to the same cluster) and cannot-link (two samples must belong to different clusters) relations. Several algorithms extend classical clustering frameworks to enforce these constraints, such as Seeded-K-means for partially labelled data (Basu et al., 2002), COPK-means for hard constraint satisfaction (Wagstaff et al., 2001), and PCK-means for soft constraint penalties (Basu et al., 2004). These approaches incorporate relational supervision while retaining flexibility in cluster discovery.

Metric-based semi-supervised clustering aims to learn a distance function that enforces consistency with labels or pairwise constraints (Soleymani Baghshah & Bagheri Shouraki, 2010). Must-link pairs are encouraged to be close, while cannot-link pairs are pushed apart, after which clustering is performed in the learned metric space (González-Almagro et al., 2025). A wide range of distance measures have been explored, including Mahalanobis metrics (Xing et al., 2002), kernelized metrics (Chatpatanasiri et al., 2010), graph-based distances (Klein et al., 2002), localized metrics (Frome et al., 2006), and domain-specific measures such as string-edit distance (Bilenko & Mooney, 2003) and KL divergence (Basu et al., 2006). Deep metric learning further extends these ideas by modeling nonlinear relationships (Chopra et al., 2005; Li et al., 2020), though often at the expense of interpretability.

Most metric-based approaches treat metric learning and clustering as separate stages, excluding unlabelled data during metric training. MPCK-means (Bilenko et al., 2004) addressed this limitation by jointly learning distance metrics and cluster assignments under pairwise constraints, later formalized within the Hidden Markov Random Field (HMRF) framework (Basu et al., 2006). Subsequent work extended this paradigm to representation- and kernel-based settings, jointly learning embeddings or transformations with clustering (Yang et al., 2015; Domeniconi et al., 2011; Hazratgholizadeh et al., 2022).

Probabilistic constrained clustering methods, often based

---

[1]School of Informatics, University of Edinburgh, Edinburgh, UK [2]Usher Institute, University of Edinburgh, Edinburgh, UK. Correspondence to: Luwei Wang <luwei.wang@ed.ac.uk>.

*Proceedings of the $43^{rd}$ International Conference on Machine Learning*, Seoul, South Korea. PMLR 306, 2026. Copyright 2026 by the author(s).

on mixture models, generalize these ideas to probabilistic frameworks, employing various distributional assumptions for clusters (Law et al., 2004; Zhao & Miller, 2005; Riverain et al., 2023). However, most approaches neglect uncertainty quantification, and only Echraibi et al. (2019) introduced Bayesian priors on model parameters for this purpose. Moreover, prior work has largely overlooked cluster-specific feature relevance, despite early attempts such as Li et al. (2008), incorporating feature saliency within constrained mixture models. Recent unsupervised work (Rao & Kirk, 2025) revisits variable selection in variational Bayesian mixtures, but without the pairwise-constraint setting we address.

Despite this progress, existing methods face key limitations in *scalability* (classical constraint-based methods require full dataset passes), *interpretability* (deep methods sacrifice transparency), and *robustness to noisy supervision* (fixed constraint weights cannot handle inconsistent labels). We propose BASIL[1] (**Ba**yesian **S**emi-superv**I**sed c**L**ustering), a unified Bayesian generative model that jointly captures HMRF-structured cluster assignments, per-cluster feature relevance, and latent constraint reliability within a single hierarchical formulation, with stochastic variational inference (SVI) providing scalable joint posterior inference.

**Contributions** (1) We extend the HMRF-based Bayesian semi-supervised clustering framework using standard must/cannot-link constraints and stochastic variational inference (SVI), enabling efficient inference on large datasets. (2) We integrate feature relevance learning into the generative process and derive efficient approximated updates for feature importance weights within our SVI framework, enabling the scalable identification of interpretable, cluster-specific discriminative features. (3) Our model adaptively learns latent constraint weights to identify and down-weight noisy or inconsistent supervisory signals, improving robustness when a proportion of pairwise constraints are erroneous. (4) We systematically evaluate on synthetic datasets, canonical digit benchmarks, and real-world large-scale health data to demonstrate applicability.

## 2. Related Works

**Constraint-based Methods** Seminal work in constrained clustering began with methods that modify the objective function to penalize constraint violations. PCK-means (Basu et al., 2004) extends the K-means objective by adding penalty terms for violating must-link and cannot-link constraints. Although effective for grouping data consistent with supervision, PCK-means relies on a fixed Euclidean distance, limiting its ability to handle complex data distributions where feature relevance varies. Addressing this, Basu et al. (2006) introduced the Hidden Markov Random

Field (HMRF) framework to unify constraint satisfaction with learnable distance metrics (e.g., MPCK-means). While effective, such approaches typically remain tied to K-means assumptions, modelling isotropic or elliptical cluster shapes.

**Metric Learning** A prevalent paradigm in metric-based clustering involves a decoupled two-stage process: (1) learning a distance metric from constraints, and (2) clustering in the transformed embedding space (Xing et al., 2002). However, this separation prevents the metric learning phase from leveraging the global distribution of unlabelled data, often leading to sub-optimal cluster boundaries (Bilenko et al., 2004).

**Deep Constrained Clustering** Deep constrained clustering methods have addressed some of these limitations by using nonlinear transformations to learn latent representations that satisfy pairwise constraints (Li et al., 2020). While recent advances attempt to better integrate unlabelled data through self-supervised representation learning (Hazratgholizadeh et al., 2022), these models remain predominantly deterministic. Manduchi et al. (2021) proposed a deep conditional Gaussian mixture model (DCGM) that combines deep generative modeling with pairwise constraints, using stochastic gradient variational Bayes. However, it does not address adaptive constraint weighting or cluster-specific feature selection. While deep models provide flexible nonlinear representations, they typically lack interpretability, which limits their applicability in domains requiring transparent decision-making.

**Bayesian Approaches** Generative Bayesian formulations provide a natural framework for modelling epistemic uncertainty, cluster-specific covariance structures, and incorporating prior knowledge. Despite these advantages, they remain underexplored in the context of constrained clustering. Existing Bayesian extensions of the HMRF framework, such as Echraibi et al. (2019), are restricted to must-link constraints and rely on traditional variational inference. These restrictions hinder the model's ability to handle complex cannot-link relationships, scale to large-scale datasets, and obtain uncertainty in constraint reliability, as they assume fixed constraint weights. A detailed comparison between BASIL and its closest baselines is provided in Appendix A.1.

## 3. Problem Formulation

### 3.1. Preliminaries

Let $X = \{\mathbf{x}_n\}_{n=1}^N$ denote a dataset of $N$ observations, where each $\mathbf{x}_n \in \mathbb{R}^D$ is a $D$-dimensional feature vector. The objective is to partition the data into $K$ clusters. Let $\mathbf{z} = \{z_n\}_{n=1}^N$ represent the set of latent cluster assignments, where $z_n \in \{1, \ldots, K\}$ denotes the cluster index of $\mathbf{x}_n$.

---

[1]Code is available at our GitHub repository.

We assume that partial prior knowledge is available in the form of pairwise constraints. Let $\mathcal{C}_+$ denote the set of must-link constraints, where a pair $(\mathbf{x}_i, \mathbf{x}_j) \in \mathcal{C}_+$ indicates that $\mathbf{x}_i$ and $\mathbf{x}_j$ should belong to the same cluster. Similarly, let $\mathcal{C}_-$ denote the set of cannot-link constraints, where $(\mathbf{x}_i, \mathbf{x}_j) \in \mathcal{C}_-$ indicates they should belong to different clusters. This supervision is encoded in a symmetric constraint matrix $Y = \{y_{ij}\}_{1 \le i < j \le N}$, where $y_{ij} = 1$ if $(\mathbf{x}_i, \mathbf{x}_j) \in \mathcal{C}_+$, $y_{ij} = -1$ if $(\mathbf{x}_i, \mathbf{x}_j) \in \mathcal{C}_-$, and $y_{ij} = 0$ for unconstrained pairs.

### 3.2. Constraint Integration via HMRFs

To incorporate these relational constraints into a probabilistic clustering framework, we model the dependencies between latent variables using a Hidden Markov Random Field (HMRF) (Basu et al., 2006). In this graph-based formulation, each latent variable $z_n$ is a node, and edges connect variables constrained by $y_{nm} \ne 0$.

For each assignment $z_n$, we define its constraint neighbors $G_n = \{z_m \mid y_{nm} \ne 0\}$ as the set of samples linked to it by a constraint. The HMRF assumes that the conditional distribution of $z_n$ given the remaining variables depends only on its constraint neighbors. By the Hammersley-Clifford theorem (Hammersley & Clifford, 1971), the joint probability of the label configuration $\mathbf{z}$ conditioned on constraints $Y$ follows a Gibbs distribution,

$$p(\mathbf{z} \mid Y) = \frac{1}{L} \prod_{n=1}^{N} \prod_{z_m \in G_n} \exp\left(-\mathcal{V}(z_n, z_m)\right), \quad (1)$$

where $L$ is the partition function and $\mathcal{V}(z_n, z_m)$ is a pairwise potential function, or energy function, that penalizes constraint violations.

Standard formulations, such as in Basu et al. (2006), define the potential function to enforce spatial consistency,

$$\mathcal{V}(z_n, z_m) =$$
$$\begin{cases} w_{nm} f(\mathbf{x}_n, \mathbf{x}_m) \mathbb{I}(z_n \ne z_m) & \text{if } y_{nm} = 1, \\ \bar{w}_{nm}(f^{\max} - f(\mathbf{x}_n, \mathbf{x}_m)) \mathbb{I}(z_n = z_m) & \text{if } y_{nm} = -1, \\ 0 & \text{otherwise}, \end{cases}$$
$$(2)$$

where $\mathbb{I}(\cdot)$ is the indicator function. Here, $f(\cdot, \cdot)$ is a distance measure (e.g., Euclidean or Mahalanobis), and $f^{\max}$ is its upper bound. The terms $w_{nm}$ and $\bar{w}_{nm}$ are fixed weights reflecting the confidence in the constraints. High penalties are assigned to must-linked pairs that are distant or cannot-linked pairs that are close. A simpler variant is the *generalized Potts* potential function (Boykov et al., 1998), which specifies the cost of violating a must-link constraint to be only $w_{nm}$ irrespective of the distance.

Recent Bayesian approaches have explored alternative potentials. For instance, Echraibi et al. (2019) define $\mathcal{V}$ based on the Kullback-Leibler (KL) divergence between varia-

tional distributions of cluster parameters,

$$\mathcal{V}(z_n, z_m) = w\, \mathbb{D}_{\text{KL}}^{\text{sym}}[q(\boldsymbol{\theta}_{z_n}) \,\|\, q(\boldsymbol{\theta}_{z_m})] \text{ if } y_{nm} = 1, \quad (3)$$

where $w$ is a fixed scalar weight. However, we find that this KL divergence between variational distributions of cluster parameters mismatches with constraint weight learning. Its magnitude grows unboundedly with sample size and penalizes differences in posterior certainty rather than cluster means, often leading to degenerate solutions during joint optimization (Appendix B.3). We therefore adopt the cluster-divergence formulation of Equation (7), defined by the symmetric KL between cluster-conditional data distributions, which ensures training stability.

## 4. BASIL

We have the following generative model of the proposed Bayesian semi-supervised mixture model with feature selection and constraint weights learning,

$$\begin{aligned} \boldsymbol{\pi} &\sim \text{Dir}(\boldsymbol{\alpha}_0), \\ w_{nm} &\sim \text{Gamma}(\boldsymbol{\lambda}_w), \\ \bar{w}_{nm} &\sim \text{Gamma}(\boldsymbol{\lambda}_{\bar{w}}), \\ \mathbf{z} \mid \boldsymbol{\pi}, W, \bar{W}, Y &\sim p(\mathbf{z} \mid \boldsymbol{\pi}, W, \bar{W}, Y), \\ \boldsymbol{\theta}_k &\sim p(\boldsymbol{\theta}_k), \\ \tilde{\theta}_{kd}^{\mu} &= \gamma_{kd}\theta_{kd}^{\mu} + (1 - \gamma_{kd})\theta_{0d}^{\mu}, \\ \mathbf{x}_n \mid z_n = k, \tilde{\boldsymbol{\theta}}_k &\sim p(\mathbf{x}_n \mid z_n = k, \tilde{\boldsymbol{\theta}}_k), \quad (4) \end{aligned}$$

where $\boldsymbol{\pi} = \{\pi_k\}_{k=1}^{K}$ are the mixture weights and $\boldsymbol{\theta}_k$ are pre-selected parameters of the $k$th component. We give Gamma priors on the constraint violation costs for must-link and cannot-link $W$ and $\bar{W}$ to also learn the constraint weights. The likelihood distribution $p(\mathbf{x}_n)$ depends on the type of random variables $X$, and $p(\boldsymbol{\theta}_k)$ is usually a conjugate prior for the likelihood function. To determine which covariates actively drive the mixture components, we incorporate feature selection in clustering (Liverani et al., 2015). We shrink each cluster's mean $\theta_{kd}^{\mu}$ toward the population-level mean $\theta_{0d}^{\mu}$ using a weight parameter $\gamma_{kd} \in (0, 1)$ with $\gamma_{kd} \sim \text{Beta}(\boldsymbol{\lambda}_{\gamma 1}, \boldsymbol{\lambda}_{\gamma 2})$. The plate diagram is in Figure 1.

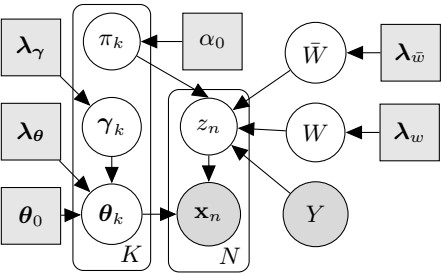

*Figure 1.* Plate diagram of BASIL with feature selection and constraint weights learning.

According to Section 3.2, the joint distribution over latent cluster assignment variables by HMRF of a semi-supervised problem is

$$p(\mathbf{z} \,|\, \boldsymbol{\pi}, W, \bar{W}, Y)$$
$$= \frac{1}{L} \prod_{n=1}^{N} \pi_{z_n}^{\mathbb{I}(G_n=\emptyset)} \prod_{z_m \in G_n} \exp(-\mathcal{V}(z_n, z_m)). \tag{5}$$

## 5. Stochastic Variation Inference for BASIL

We adopt the stochastic variational inference (SVI) (Hoffman et al., 2013) to perform inference on the model. SVI extends mean-field variational inference (Blei & Jordan, 2006) with a stochastic optimization algorithm that scales to large datasets via mini-batch learning.

The optimization objective is formalized in the following definition.

**Definition 5.1** (SVI Objective). The Evidence Lower Bound (ELBO) $\mathcal{L}(q)$ (Blei et al., 2017), which serves as the maximization objective of the model, is defined as

$$\mathcal{L}(q) = \mathbb{E}_q[\log p(X, \mathbf{z}, \boldsymbol{\Theta} \,|\, Y)] - \mathbb{E}_q[\log q(\mathbf{z}, \boldsymbol{\Theta})], \quad (6)$$

where $\boldsymbol{\Theta} = \{\boldsymbol{\pi}, W, \bar{W}, \{\boldsymbol{\theta}_k\}_{k=1}^K, \{\boldsymbol{\gamma}_k\}_{k=1}^K\}$ comprises the parameters and $q(\cdot)$ denotes the variational distributions approximating the posterior. The joint probability factorization is provided in Appendix C.

**Definition 5.2** (Constraint Potential Function). Following Basu et al. (2006), we define the adaptive pairwise potential function (cluster divergence) $\mathcal{V}(z_n, z_m)$ as:

$$\mathcal{V}(z_n, z_m) =$$
$$\begin{cases} w_{nm} \mathbb{D}_{\mathrm{KL}}^{\mathrm{sym}}[p(\mathbf{x} \,|\, \boldsymbol{\theta}_{z_n}) \,\|\, p(\mathbf{x} \,|\, \boldsymbol{\theta}_{z_m})] & \text{if } y_{nm} = 1 \\ \bar{w}_{nm} \mathcal{V}^{\mathrm{max}} \mathbb{I}(z_n = z_m) & \text{if } y_{nm} = -1 \\ 0 & \text{otherwise.} \end{cases} \tag{7}$$

In SVI, we maximize the ELBO by iteratively updating local variational parameters via closed-form coordinate ascent and performing stochastic natural gradient steps on global parameters (general SVI update rule in Appendix C). We summarize the specific update rules derived for our model in the following propositions.

**Proposition 5.3** (Local Update Rules). *The variational parameters for local latent variables are updated as follows:*

1. *Local Cluster Assignments* $\Phi_{nk}$:

$$\Phi_{nk} \propto \exp \left\{ \mathbb{E}_q[\log p(\mathbf{x}_n | z_n = k, \tilde{\boldsymbol{\theta}}_k)] \right. $$
$$+ \mathbb{I}(G_n = \emptyset) \mathbb{E}_q[\log \pi_k]$$
$$\left. - \sum_{m \in G_n} \sum_{j=1}^K \Phi_{mj} \mathcal{V}(k, j) \right\}, \quad (8)$$

where for must-links ($y_{nm} = 1$), we define $\mathcal{V}(k, j) = \mathcal{V}_m(k, j) = w_{nm} \mathbb{D}_{KL}^{sym}[p(\mathbf{x} \,|\, \boldsymbol{\theta}_k) \,\|\, p(\mathbf{x} \,|\, \boldsymbol{\theta}_j)]$, while for cannot-links ($y_{nm} = -1$), we set $\mathcal{V}(k, j) = \mathcal{V}_c(k, j) = \bar{w}_{nm} \mathrm{diag}(\mathcal{V}^{max})$. In the generalized Potts setting, this simplifies to $\mathcal{V}_m(k, j) = w_{nm}$ and $\mathcal{V}_c(k, j) = \bar{w}_{nm}$.

2. *Constraint Weights* $w_{nm}, \bar{w}_{nm}$: *The rate parameters for the Gamma posteriors are updated as:*

$$\beta_{nm,2} = \boldsymbol{\lambda}_{w2} + \mathbb{I}(m \in G_n, y_{nm} = 1)$$
$$\times \sum_{j,k} \Phi_{nj} \Phi_{mk} \mathbb{D}_{KL}^{sym},$$
$$\bar{\beta}_{nm,2} = \boldsymbol{\lambda}_{\bar{w}2} + \mathbb{I}(m \in G_n, y_{nm} = -1)$$
$$\times \sum_{k} \Phi_{nk} \Phi_{mk} \mathcal{V}^{max}. \tag{9}$$

**Proposition 5.4** (Global Update Rules (SVI Steps)). *The global variational parameters are updated using the stochastic formula derived from coordinate ascent:*

1. *Feature Importance* $\gamma_{kd}$. *The first-order expectation of $\log \tilde{\theta}_{kd}$ yields a tractable surrogate $\tilde{\mathcal{L}}$ to $\mathcal{L}(q)$ (Appendix D) and a non-conjugate factor that admits a conjugate-computation variational inference (CVI) interpretation (Khan & Lin, 2017), with the relevance score $\hat{l}_{kd}$ as its pseudo-sufficient-statistics. The parameters $c_{kd}, d_{kd}$ are then approximately updated via mini-batch CVI,*

$$\hat{c}_{kd} = \boldsymbol{\lambda}_{\gamma 1} + \frac{N}{M} \max(0, \hat{l}_{kd}),$$
$$\hat{d}_{kd} = \boldsymbol{\lambda}_{\gamma 2} + \frac{N}{M} \max(0, -\hat{l}_{kd}), \tag{10}$$

*where $\hat{l}_{kd}$ is the mini-batch relevance score. The nonnegative projection $\max(0, \cdot)$ is sign-aligned with the proper CVI natural gradient (since $d_{kd}/(c_{kd}+d_{kd})^2 > 0$), so each iterate ascends $\tilde{\mathcal{L}}$ in expectation and the limit point lies within $O(\boldsymbol{\lambda}_{\gamma 2}^2/(\boldsymbol{\lambda}_{\gamma 1} + |\hat{l}_{kd}|)^3)$ of a stationary point of $\tilde{\mathcal{L}}$ (Appendix D).*

2. *Mixture Parameters $\boldsymbol{\theta}_k$ (Binary Data)*: *For multivariate Bernoulli data, parameters $a_{kd}, b_{kd}$ of Beta distribution are updated by applying the SVI step to the intermediate coordinate ascent values weighted by expected feature relevance:*

$$\hat{a}_{kd} = \boldsymbol{\lambda}_{\theta 1} + \mathbb{E}_q[\gamma_{kd}] \frac{N}{M} \sum_{n \in \mathcal{B}_t} \Phi_{nk} x_{nd},$$
$$\hat{b}_{kd} = \boldsymbol{\lambda}_{\theta 2} + \mathbb{E}_q[\gamma_{kd}] \frac{N}{M} \sum_{n \in \mathcal{B}_t} \Phi_{nk}(1 - x_{nd}). \tag{11}$$

Note that the update rule for the mixture weights $\boldsymbol{\pi}$ follows the standard Dirichlet posterior update and is thus omitted

here. The detailed derivation of update rules can be found in Appendix C. The outline of the algorithm is presented in Algorithm 1.

---

**Algorithm 1** Stochastic Variational Inference for BASIL

---

**Require:** Data $X \in \{0,1\}^{N \times D}$, clusters $K$, constraints $Y$, batch size $M$
**Ensure:** Variational parameters $\{\boldsymbol{\nu}, \Phi\}$ maximizing the ELBO

1: **Init:** Construct constraint graphs $G^+$ and $G^-$ from $Y$; initialize $\Phi$ via K-medoids; set priors
2: **for** $t = 1, 2, \ldots$ until convergence **do**
3:     Sample mini-batch $\mathcal{B}_t$ of size $M$
4:     $\rho_t \leftarrow (\tau + t)^{-\kappa}$                 ▷ *Step size*
5:     *// Global update: Mixtures*
6:     Compute $\hat{a}_{kd}, \hat{b}_{kd}$ via Equation (11)
7:     Compute $\hat{\alpha}_k \leftarrow \alpha_0 + \frac{N}{M} \sum_{m=1}^{M} \Phi_{mk}$
8:     $\boldsymbol{\nu}^{(t)} \leftarrow (1 - \rho_t)\boldsymbol{\nu}^{(t-1)} + \rho_t \hat{\boldsymbol{\nu}}$         ▷ *SVI update*
9:     *// Local update:*
10:    **if** KL potential enabled **then**
11:        Compute $\mathcal{V}(k, j)$ via Equation (7)
12:    **else**
13:        Reduce $\mathcal{V}(k, j)$ to *generalized Potts* form
14:    **end if**
15:    **if** adaptive weights enabled **then**
16:        Update $\mathbb{E}[w_{nm}], \mathbb{E}[\bar{w}_{nm}]$ via Equation (9)
17:    **end if**
18:    **for** $m \in \mathcal{B}_t$ **do**
19:        Update $\Phi_{mk}$ via Equation (8)     ▷ *Cluster assign*
20:    **end for**
21:    **if** feature selection enabled **then**
22:        Compute $\hat{c}_{kd}, \hat{d}_{kd}$ via Equation (10) and update
                   ▷ *Feature importance*
23:        $\mathbb{E}[\gamma_{kd}] \leftarrow \frac{c_{kd}}{c_{kd} + d_{kd}}$
24:    **end if**
25: **end for**

---

**Computational Complexity** The per-batch cost of BASIL is $O(MKD) + O(M g_{\max} K)$, where $g_{\max}$ the maximum number of constraint neighbours per sample. Which term dominates depends on the supervision and feature regimes. Under dense supervision ($g_{\max} \gg D$), the constraint dominates. Under high-dimensional sparse supervision ($D \gg g_{\max}$), the feature dimension dominates and the per-batch matmul cost grows linearly in $D$. Feature selection contributes an additional $O(MKD)$ term of the same form as the base mixture update, roughly doubling the matmul cost without changing the scaling order.

# 6. Experiments

To evaluate the performance of our proposed model, we first conducted a sanity check using a synthetic binary dataset to validate our implementation (detailed results and analysis are provided in Appendix B.1). We then benchmarked the model against two standard handwritten digit datasets: DIGITS and MNIST. We choose these datasets because they are grayscale digit images that can be binarized effectively, allowing us to test the model's capability on real-world data with less information loss. Finally, we applied the model to real-world large-scale health datasets to illustrate its applicability.

## 6.1. Empirical Evaluation

**Experimental Setup** We benchmark against metric-based (MML, Xing et al. (2002)), constraint-based (PCK, Basu et al. (2004)), hybrid (MPCK, Basu et al. (2006)), and deep generative (DCGM, Manduchi et al. (2021), FC backbone 128-64-32-10) baselines under 0%, 20%, and 50% supervision levels. Pairwise constraints are randomly sampled from ground truth labels (up to 5M pairs) and applied identically across all methods. We report Normalized Mutual Information (NMI) and Clustering Accuracy (ACC); formal definitions are provided in Appendix A.4. Timing reported using a single AMD EPYC 7742 core.

**Benchmark on Real Digit Images** The DIGITS dataset (Alpaydin & Alimoglu, 1996) contains 1,797 samples ($8 \times 8$ images, 64 dimensions) across 10 classes with an 80-20 train-test split ($N_{\text{train}} = 1,437$). The MNIST dataset (LeCun et al., 2010) comprises 70,000 $28 \times 28$ images (784 dimensions) with the standard 60k/10k split. Both datasets are binarized as described in Appendix A.2.

**Scalability** BASIL demonstrates superior scalability compared to baselines (Table 1). On the DIGITS dataset, it maintains competitive clustering accuracy even with limited supervision (0 to 0.03 improvement). Crucially, on the larger MNIST benchmark ($N$=60,000), BASIL leverages SVI with a minibatch size of 500 to complete training in 18.68 minutes (unsupervised) to 1.57 hours (50% supervision). In contrast, all baseline methods (MML, PCK-means, MPCK-means) fail to converge within 2 days, representing a **reduction of over** 96% in computation time. This validates the efficacy of our Bayesian mini-batch formulation for large-scale tasks. As predicted by the complexity analysis, the constraint potential dominates wall-clock cost under dense supervision. On DIGITS ($D$=64, $g_{\max} \approx 358$ at 50% supervision), training takes 44.4 min versus 5.8 s unsupervised, a $\sim 460\times$ increase driven purely by the constraint term. The deep baseline DCGM on MNIST requires tuning the BCE constraint penalty to $\alpha = 100$ (the default $\alpha = 10000$), yet roughly half the seeds still fail at 20% supervision ($0.51 \pm 0.13$). At 50% supervision, DCGM reaches $0.72 \pm 0.07$ versus BASIL $0.79 \pm 0.00$ at roughly twice the runtime. On DIGITS, DCGM matches classical

*Table 1.* Clustering Accuracy and Time of Models Aggregated over 10 Runs per Image Dataset and the Same Test Dataset.

| | LABEL% | 0% | | | 20% | | | 50% | | |
|---|---|---|---|---|---|---|---|---|---|---|
| DATA | MODEL | TRAIN | TEST | TIME | TRAIN | TEST | TIME | TRAIN | TEST | TIME |
| DIGITS | MML | – | – | – | $0.57\pm0.02$ | $0.61\pm0.03$ | $1.10\,\text{m}\pm0.02$ | $0.62\pm0.03$ | $0.71\pm0.05$ | $4.18\,\text{m}\pm0.04$ |
| | PCK | – | – | – | $0.85\pm0.00$ | $\mathbf{0.80}\pm0.01$ | $9.92\,\text{s}\pm1.91$ | $0.92\pm0.00$ | $0.84\pm0.00$ | $4.70\,\text{m}\pm0.80$ |
| | MPCK | – | – | – | $0.75\pm0.05$ | $0.75\pm0.02$ | $1.49\,\text{m}\pm0.02$ | $0.80\pm0.04$ | $0.76\pm0.07$ | $9.68\,\text{m}\pm0.04$ |
| | DCGM | $0.13\pm0.02$ | $0.16\pm0.02$ | $1.19\,\text{m}\pm0.06$ | $0.51\pm0.30$ | $0.51\pm0.30$ | $2.18\,\text{m}\pm0.03$ | $0.62\pm0.38$ | $0.61\pm0.38$ | $2.19\,\text{m}\pm0.03$ |
| | BASIL | $0.69\pm0.02$ | $0.68\pm0.01$ | $5.81\,\text{s}\pm0.34$ | $0.83\pm0.02$ | $\mathbf{0.80}\pm0.02$ | $7.00\,\text{m}\pm0.25$ | $0.93\pm0.00$ | $\mathbf{0.87}\pm0.00$ | $44.4\,\text{m}\pm1.20$ |
| MNIST | MML | – | – | – | – | – | $>2\,\text{d}$ | – | – | $>2\,\text{d}$ |
| | PCK | – | – | – | – | – | $>2\,\text{d}$ | – | – | $>2\,\text{d}^{\dagger}$ |
| | MPCK | – | – | – | – | – | $>2\,\text{d}$ | – | – | $>2\,\text{d}^{\dagger}$ |
| | DCGM$^{\ddagger}$ | $0.11\pm0.00$ | $0.11\pm0.00$ | $1.73\,\text{h}\pm0.04$ | $0.51\pm0.13$ | $0.51\pm0.13$ | $1.95\,\text{h}\pm0.39$ | $0.72\pm0.07$ | $0.72\pm0.07$ | $2.54\,\text{h}\pm0.79$ |
| | BASIL | $0.58\pm0.02$ | $0.59\pm0.02$ | $18.68\,\text{m}\pm0.41$ | $0.73\pm0.03$ | $\mathbf{0.68}\pm0.02$ | $1.52\,\text{h}\pm0.02$ | $0.89\pm0.00$ | $\mathbf{0.79}\pm0.00$ | $1.57\,\text{h}\pm0.01$ |
| CHMNIST | BASIL | $0.52\pm0.01$ | $0.52\pm0.01$ | $0.54\,\text{h}\pm0.01$ | – | – | – | $0.84\pm0.01$ | $0.68\pm0.01$ | $1.04\,\text{h}\pm0.01$ |

$^{\dagger}$ LARGE MEMORY DEMAND. $^{\ddagger}$ DCGM ON MNIST REQUIRES TUNING THE BCE CONSTRAINT PENALTY TO $\alpha = 100$ (VERSUS THE DEFAULT $\alpha = 10000$) TO MITIGATE COLLAPSE.

baselines in mean accuracy but exhibits substantially higher run-to-run variance (SD $\approx 0.30$ vs. $\leq 0.07$ for BASIL).

**Interpretability** A key advantage of BASIL is its capacity to explicitly model feature weights, providing superior interpretability over traditional baselines. BASIL jointly infers cluster-specific feature relevance parameters as part of the probabilistic generative process. This explicit modelling allows for granular insight into which features drive the formation of specific clusters. Qualitative analysis on MNIST demonstrates this capability, where high-importance features spatially align with discriminative digit strokes for each cluster, and Figure 2 shows feature importance maps for all digits. Quantitatively, we use a synthetic feature-selection benchmark partitioning features into informative, irrelevant, and noninformative groups. Treating $\mathbb{E}_q[\gamma_{kd}]$ as a ranking score yields **AUROC** $= 1.00$ uniformly across $N \in \{500, 5,000, 50,000\}$, averaged over $K$ recovered clusters and 10 random seeds (Appendix B.2). Furthermore, this integrated feature selection correlates with faster convergence rates, as the optimization landscape simplifies by focusing on informative dimensions. In contrast, none of the baselines (MML, PCK, MPCK, DCGM) provide explicit, probabilistic per-cluster feature attribution, limiting their applicability in domains where transparent decision-making matters.

**Adaptive Constraint Weighting** A key innovation of BASIL is its ability to robustly handle noisy supervision through its fully Bayesian formulation, in contrast to baselines such as PCK-means and MPCK-means which rely on fixed penalty parameters. By modelling constraint reliability as a latent variable with a Gamma prior, BASIL adaptively down-weights inconsistent constraints during inference. The Gamma prior hyperparameters $\boldsymbol{\lambda}_w = (\boldsymbol{\lambda}_{w1}, \boldsymbol{\lambda}_{w2})$ control the expected constraint weight. We use $\text{Gamma}(1, 1)$ as the default weakly informative prior with unit mean. Increasing the rate parameter $\boldsymbol{\lambda}_{w2}$ reduces the prior mean to encourage stronger down-weighting. In practice, when constraints are reliable, fixing $w_{nm}$ avoids optimisation complexity, and when noise is suspected, adaptive weights allow the model to learn which constraints to trust. To quantitatively validate the latter, with 20% corrupted constraints on DIGITS at 20% supervision the posterior mean $\mathbb{E}_q[w_{nm}]$ separates clean from corrupted pairs with **AUROC** $= 0.97 \pm 0.01$ (and $0.93 \pm 0.01$ at 30% noise, Appendix B.2). The mechanism detects constraint *violation* relative to the current cluster geometry rather than noise per se, yet does not penalise valid hard constraints (Appendix B.2). Experimental results confirm that BASIL prevents performance degradation in low-noise regimes (Figure 3) where deterministic baselines struggle, and reliably outperforms competitors as noise increases. BASIL maintains robustness even at 30% noisy supervision. By contrast, all baselines degrade as noise rises, most sharply DCGM which collapses from 0.74 at 20% to 0.21 at 50% noise at 20% supervision (Appendix Tables B.17 to B.20).

**Ablation Study** We conduct a comprehensive ablation study (Appendix B.3) to isolate the contributions of stochastic variational inference (SVI), the cluster-divergence KL potential, adaptive constraint weighting, and feature selection. SVI delivers the scalability gain, reaching NMI $= 0.77$ in 19.42 min on $N{=}50,000$ synthetic data at 50% supervision versus 18.95 hr (NMI 0.80) for the closest full-batch Bayesian baseline (Tables B.5 and B.6), a $\sim 58\times$ wall-clock reduction with comparable test NMI. Results further show that the cluster-divergence KL potential guides how the model enforces supervision, jointly optimising cluster structure and feature weights to best fit constraints while reducing sensitivity to hyperparameter initialisation (Figure B.3). Feature selection accelerates convergence without hurting

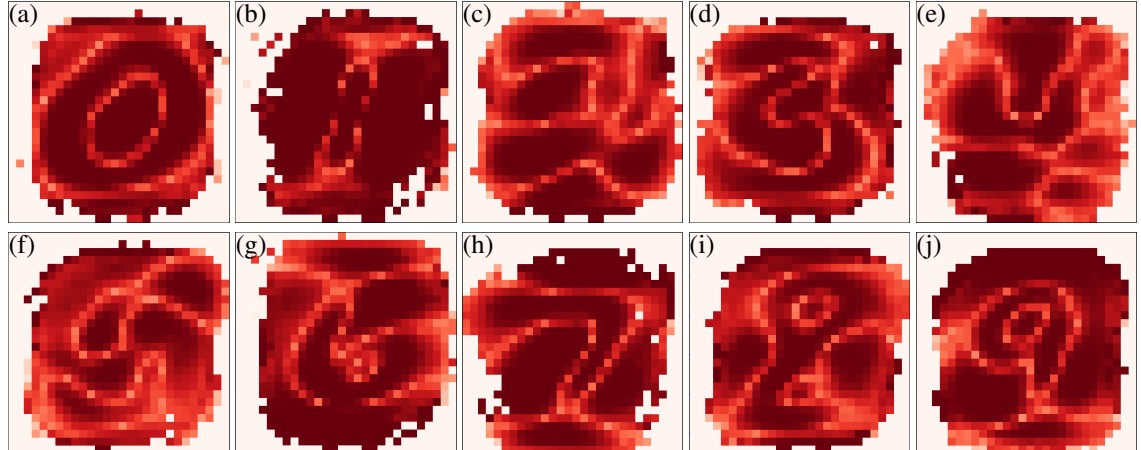

*Figure 2.* Feature importance heatmaps learned by BASIL on the MNIST dataset. (a)-(j) are for digits from 0-9. Darker color implies more importance.

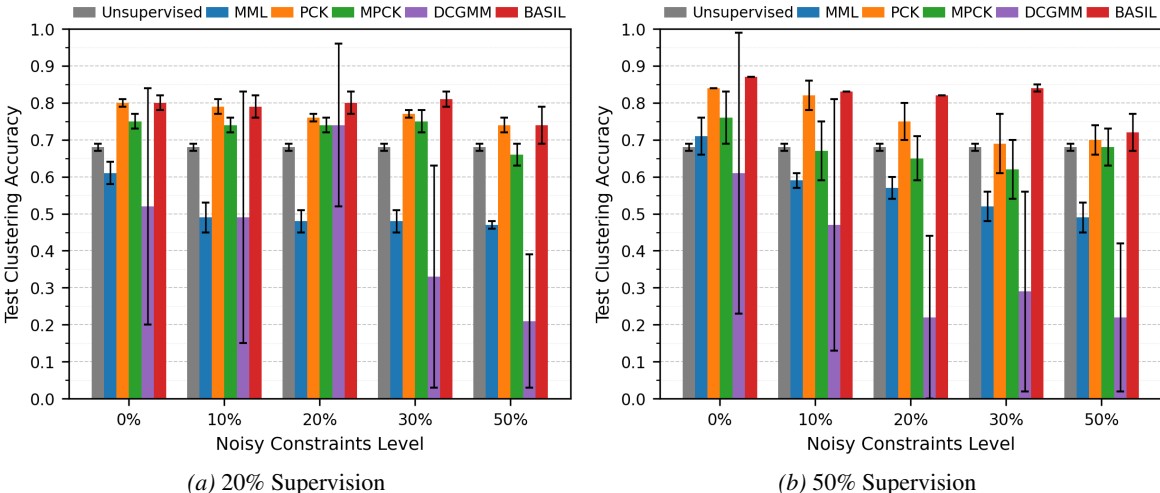

*(a)* 20% Supervision

*(b)* 50% Supervision

*Figure 3.* Test clustering accuracy on DIGITS dataset across different noisy-constraint levels for different models.

accuracy and interacts strongly with adaptive constraint weighting, and only their combination tolerates heavy noise, gaining 4 ACC points at 20% noisy constraints and retaining ACC = 0.74 at 50% noise (Table B.23). However, jointly optimising the cluster-divergence KL potential with adaptive constraint weights can become unstable (Appendix B.3). We therefore employ a phased optimisation strategy where an initial warmup phase controlled by warmup proportion uses fixed constraint weights with the cluster-divergence KL potential and feature selection to stabilise the feature manifold, after which the model transitions to adaptive weight inference to down-weight noisy constraints.

### 6.2. Application to Health Data

**ChestMNIST**  We evaluate BASIL on the CHMNIST dataset (Yang et al., 2023), comprising 112,120 chest X-ray images ($28 \times 28$, 784 dimensions) with 14 disease labels.

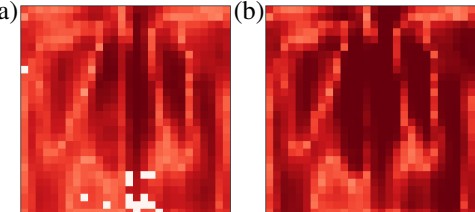

*Figure 4.* Feature importance heatmaps learned by BASIL on the CHMNIST dataset. (a) No cardiomegaly and (b) cardiomegaly. Darker color implies more importance.

We convert it to binary clustering by selecting cardiomegaly as the target, motivated by clinical relevance and suitability for pixel-level feature importance interpretation. Images are binarized using adaptive thresholding (Appendix A.2).

On CHMNIST, BASIL achieves test accuracy of 0.68 with 50% supervision, with training accuracy reaching 0.84 (Ta-

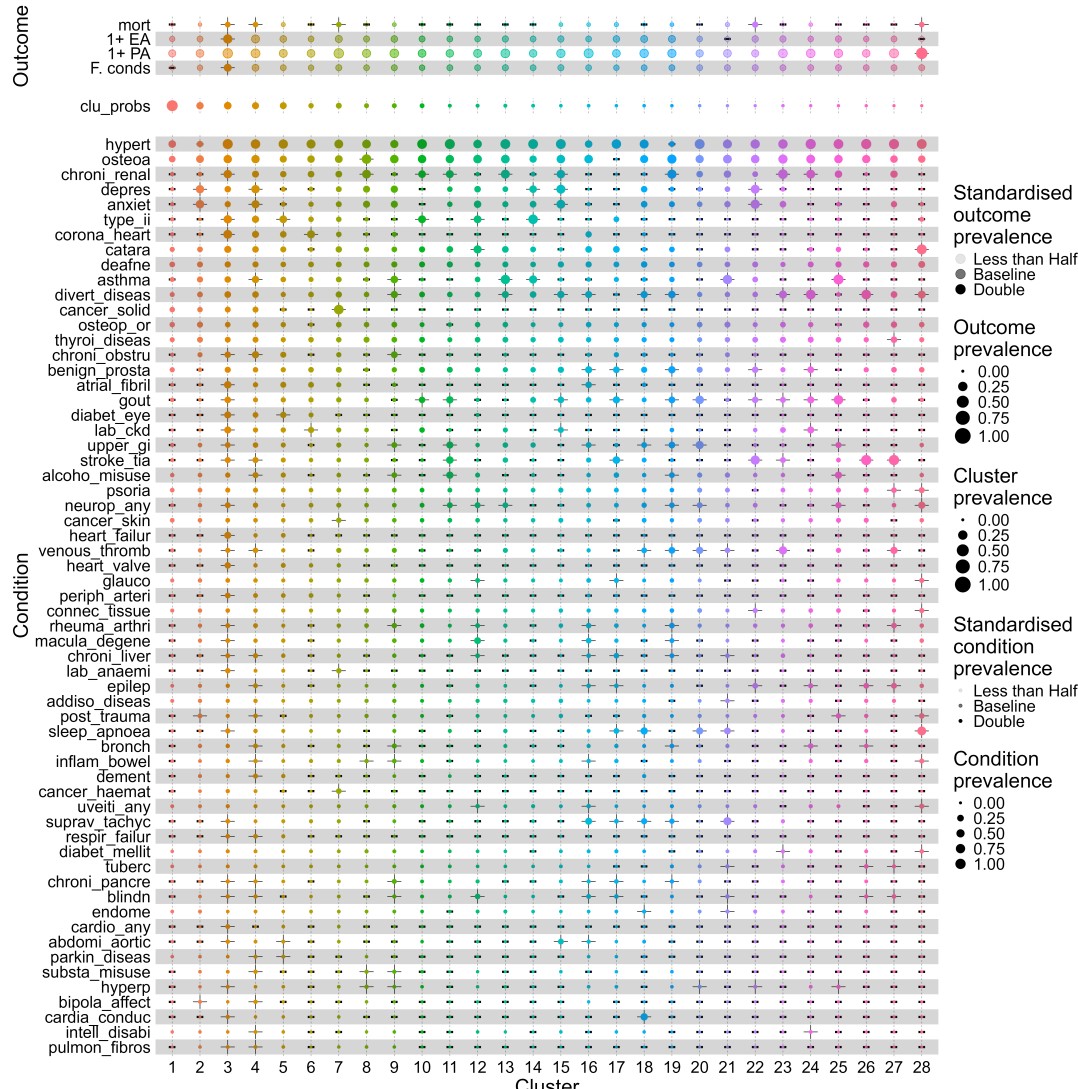

*Figure 5.* Cluster profiles learned by BASIL on the CPRD dataset. Bubble size indicates condition prevalence within each cluster; $+/-$ symbols highlight conditions or outcomes with relative risk $> 2\times$ or $< 0.5\times$ the population rate. Clusters are ordered based on their sizes (clu_probs) and conditions are ordered based on their respective marginal prevalences.

ble 1). The train-test gap and moderate overall performance can be attributed to information loss introduced by the binarization preprocessing. Nevertheless, the feature importance maps (Figure 4) reveal clinically interpretable patterns: the model assigns high importance (dark regions) to the cardiac silhouette, with the cardiomegaly class exhibiting a wider high-importance region corresponding to an enlarged heart, mirroring how radiologists assess cardiomegaly via the cardiothoracic ratio.

**CPRD** We further evaluate BASIL on a subset of the Clinical Practice Research Datalink (CPRD) (Herrett et al., 2015), a UK primary care database, with individuals aged 40 years or older as of 1 January 2018 with at least one of 83

pre-selected conditions. We focus on the 70–74 age group ($N = 501, 207$ after excluding conditions with prevalence $< 0.5\%$), represented by 61 binary condition indicators with a stratified 50-25-25 train-validation-test split. Outcomes are defined over a 1-year follow-up as an 8-level categorical variable combining three binary adverse events: planned hospital admission (PA), emergency admission (EA), and mortality (mort). Must-link constraints imply that two individuals have the same outcomes and cannot-link imply they differ in at least one. We also report the (empirical) average number of conditions in the clusters (F. conds).

The number of clusters, constraint weight learning settings, and constraint sampling schemes were chosen based on their alignment to outcomes (measured as the number of clusters

with $> 2\times$ and $< 0.5\times$ of outcome prevalence compared to sub-population average, higher is better), and diversity (measured as the average Chebyshev similarity among cluster representations, higher is better) within the validation set (Figure B.5), together with clustering stability (Figure B.6). We show the solution for 28 clusters, weight learning under $\boldsymbol{\lambda}_{w2} = 1$, and constraint sampling that balances among the outcome classes in Figure 5 (see Section B.4 for further details).

The clustering reveals clinically interpretable patient subgroups related to outcomes; for example, cluster 3 is associated with higher mortality and emergency admission and has higher multimorbidity burden (in terms of number of conditions). Cluster 28, on the other hand, has higher planned admission and mortality but lower emergency admission while cluster 1 has lower mortality and multimorbidity burden. In general, the inferred clusters show greater alignment to outcomes, and greater diversity compared to clusters inferred from unsupervised clustering (Figure B.6).

## 7. Discussion

We presented BASIL, a scalable Bayesian framework for semi-supervised clustering that jointly addresses feature selection and adaptive constraint weighting. Our experimental results demonstrate that the proposed approach achieves competitive clustering performance while substantially improving scalability, interpretability and robustness over existing baselines.

**Scalability and Practical Efficiency**    SVI enables BASIL to scale to large datasets where traditional methods become computationally prohibitive. On MNIST, our approach completes training within hours, whereas PCK and MPCK fail to converge within two days. This advantage is particularly relevant for real-world health applications such as CHMNIST (over 100,000 chest X-ray images) and the CPRD cohort (over 500,000 patients).

**Interpretability Through Feature Selection**    The learned feature weights semantically align with discriminative digit strokes (Figure 2), offering interpretability inaccessible to traditional baselines. In high-stakes domains such as healthcare, BASIL explicitly reveals *why* patients are grouped by identifying which features (e.g., diagnostic biomarkers or anatomical structures) drive cluster formation. Such transparency builds clinician trust, enables validation against domain expertise, and can uncover disease subtypes hidden in opaque clusterings.

**Robustness to Noisy Supervision**    Adaptive constraint weighting maintains performance under up to $30\%$ noisy constraints with sufficient supervision. This is achieved through the Bayesian treatment of constraint weights as latent variables with Gamma priors, enabling principled downweighting of inconsistent constraints during inference.

**Limitations**    Our current benchmarks use a multivariate Bernoulli likelihood, with the Gaussian extension outlined in Appendix A.5. The variational family assumes mean-field conditional independence across features given the cluster assignment, which may fail to capture spatial correlations between neighbouring pixels in image data, and structured variational families could address this at increased compute cost. Adaptive constraint weighting introduces moderate run-to-run variability via the Gamma posterior, so the prior $(\boldsymbol{\lambda}_{w1}, \boldsymbol{\lambda}_{w2})$ requires light tuning to balance flexibility with stability. Jointly optimising the cluster-divergence KL potential with adaptive weighting can become unstable, which the phased schedule mitigates by first stabilising cluster geometry under fixed weights before enabling weight inference. The current single-core NumPy implementation also makes $D \gtrsim 10^4$ impractical, although this is purely an implementation bottleneck rather than an algorithmic one.

**Practical Recommendations**    The two knobs that most affect performance are the constraint-weight mode and the warmup proportion. *Trusted constraints.* Fix $w_{nm}=1$ and combine the cluster-divergence KL potential with feature selection, where the cluster-divergence KL potential sharpens cluster boundaries without optimisation variability. *Noisy constraints.* Use adaptive weights $w_{nm} \sim \mathrm{Gamma}(1, \boldsymbol{\lambda}_{w2})$ with $\boldsymbol{\lambda}_{w2} \in [1.5, 5]$, drop the cluster-divergence KL potential, and keep feature selection under the Potts potential, since joint cluster-divergence KL plus adaptive weighting can destabilise. *Unknown noise.* The warmup_prop schedule (Table A.2) runs the fixed-weight configuration for the first warmup_prop fraction of iterations before switching to adaptive weighting. This heuristic initialisation is not covered by Theorem D.6 and a theoretical treatment is left to future work (Appendix D). *High-dimensional inputs* $(D \gtrsim 10^4)$. Downsample (e.g. $56\times56$, $D=3{,}136$) and port the dominant $O(MKD)$ operations $(\Phi^\top X)$ to a GPU framework such as JAX or PyTorch, which is expected to yield a $10$–$50\times$ speedup without changing the algorithm. Default values for all other hyperparameters (Table A.2), including the feature-weight prior, are robust across our benchmarks.

**Future Directions**    The most immediate extension is to a Gaussian likelihood, outlined in Appendix A.5. Synthetic-Gaussian results in Appendix B.1 (Figure B.1) suggest the family is well-behaved under semi-supervision, and benchmarking the full BASIL Gaussian path on continuous real-world datasets is an immediate next step. Non-mean-field variational families could further improve accuracy and scalability beyond our mean-field convergence guarantees (Appendix D).

## Acknowledgements

This study is based in part on data from the Clinical Practice Research Datalink (CPRD). The study was approved by the CPRD Independent Scientific Advisory Committee (ISAC), protocol 21_000542. Wang and Wang are supported by the United Kingdom Research and Innovation (grant EP/S02431X/1), UKRI Centre for Doctoral Training in Biomedical AI at the University of Edinburgh, School of Informatics. Panas, Guthrie and Seth are partly supported by the National Institute for Health and Care Research (NIHR) under its Artificial Intelligence for Multiple and Long-Term Conditions Programme (reference number NIHR202639). The views expressed are those of the author and not necessarily those of the NIHR or the Department of Health and Social Care. Panas, Guthrie and Seth are partly supported by the Legal & General Group (research grant to establish the independent Advanced Care Research Centre at the University of Edinburgh). The funder had no role in the conduct of the study, interpretation or the decision to submit for publication. The views expressed are those of the authors and not necessarily those of Legal & General.

## Impact Statement

This paper presents foundational research in Bayesian semi-supervised clustering methodology. The proposed approach has broad potential applications in domains such as healthcare (e.g., patient stratification), biology (e.g., gene expression analysis), and social sciences (e.g., survey data analysis), where incorporating domain expertise through pairwise constraints can improve clustering quality.

As with any clustering methodology, there are potential concerns regarding misuse in surveillance or discriminatory profiling applications. However, our work does not introduce new capabilities beyond existing clustering methods; rather, it improves the statistical rigour and scalability of incorporating prior knowledge into clustering. We encourage practitioners to apply this methodology responsibly and in accordance with ethical guidelines relevant to their application domains.

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

# A. Additional Model Implementation Details

## A.1. Position of BASIL Relative to Existing Methods

BASIL formulates HMRF-structured cluster assignments, mixture likelihoods, per-cluster feature relevance, and latent constraint reliability as random variables within a single hierarchical generative model, with SVI delivering joint posterior inference. While each ingredient has appeared separately in earlier work, this unified Bayesian treatment under SVI is novel in the following four respects. First, prior HMRF-based approaches (Basu et al., 2006; Echraibi et al., 2019) treat constraint weights as fixed hyperparameters, whereas we promote them to latent variables with Gamma priors and derive closed-form local updates compatible with mini-batching. Second, feature relevance in constrained clustering has been studied only with non-Bayesian penalties (Li et al., 2008), while our Beta-shrinkage formulation $\tilde{\theta}_{kd} = \gamma_{kd}\theta_{kd} + (1 - \gamma_{kd})\theta_{0d}$ produces interpretable per-cluster relevance scores with a CVI-compatible update rule. Third, existing Bayesian semi-supervised clustering methods are limited to must-link constraints, and we instead handle both must- and cannot-link relations within a single HMRF energy. Fourth, the resulting framework scales to $N \geq 5 \cdot 10^5$ via SVI, two orders of magnitude beyond Echraibi et al. (2019). Table A.1 summarises these differences against the closest baselines. Empirically (Section 6.1), these design choices translate into measurable advantages in noise robustness, interpretability, and scalability that none of the individual components provide on their own.

*Table A.1.* Feature comparison of BASIL against closest semi-supervised clustering baselines. ML/CL denote must-link / cannot-link.

| Method | Constr. | Adapt. $w$ | Feat. sel. | Scalable | Uncert. |
|---|---|---|---|---|---|
| PCK (Basu et al., 2004) | ML/CL | $\times$ | $\times$ | $\times$ | $\times$ |
| MPCK (Basu et al., 2006) | ML/CL | $\times$ | $\times$ | $\times$ | $\times$ |
| SS-DPGMM (Echraibi et al., 2019) | ML | $\times$ | $\times$ | $\times$ | $\checkmark$ |
| DCGM (Manduchi et al., 2021) | ML/CL | $\times$ | $\times$ | $\checkmark$ | $\times$ |
| BASIL (ours) | ML/CL | $\checkmark$ | $\checkmark$ | $\checkmark$ | $\checkmark$ |

## A.2. Data Preprocessing

For image datasets, we apply binarization to convert grayscale images to binary format compatible with our Bernoulli likelihood model. For the DIGITS and MNIST datasets, pixel intensities are normalized to $[0, 1]$ and binarized using a fixed threshold of 0.5.

For medical imaging data (CHMNIST), we employ a two-stage adaptive binarization pipeline to preserve clinically relevant structural features: (1) Contrast Limited Adaptive Histogram Equalization (CLAHE) (Zuiderveld, 1994) enhances local contrast using a kernel size of 4 pixels to accentuate anatomical boundaries; (2) Otsu's thresholding (Otsu, 1979) determines an optimal binarization threshold per image. This approach preserves structural features that would otherwise be lost with simple fixed-threshold binarization.

## A.3. Model Hyperparameters

Table A.2 lists the default hyperparameter values used in all experiments unless otherwise noted. We set noninformative priors to all parameters except for feature weights $\gamma_{kd}$ and the constraint weights $w_{nm}, \bar{w}_{nm}$ (which use a weakly informative unit-mean Gamma prior). We initialise the probability matrix $\Phi_{nk}$ for the variational distribution by K-medoids when $N < 1,000$ or K-modes otherwise for faster speed.

We fix the forgetting rate of SVI to $\kappa = 0.7$. We check the convergence by the exponential moving average (EMA) of the relative change of parameters with the following equation:

$$\Delta^{(t)} = \frac{\sqrt[4]{N}}{2B^2} \sum_b \frac{\left\| \boldsymbol{\Theta}_b^{(t)} - \boldsymbol{\Theta}_b^{(t-1)} \right\|_2}{D_b \left\| \boldsymbol{\Theta}_b^{(t-1)} \right\|_2} \tag{A.12}$$

where $B$ is the number of variational parameters in the model and $D_b$ is the dimension of that variational parameter. Then $\Delta_{\text{EMA}}^{(t)} = 0.8\Delta_{\text{EMA}}^{(t-1)} + (1 - 0.8)\Delta^{(t)}$. This relative change is smoother and invariant to the sample size scaling law of variances for variational parameters. We set the relative tolerance to $1e^{-5}$ by default. It is recommended to tune the relative tolerance for different sample sizes to avoid underfitting by inspecting the relative change of parameters over iterations.

*Table A.2.* Default hyperparameters for BASIL.

| SYMBOL | DEFAULT | MEANING |
|---|---|---|
| $\boldsymbol{\alpha}_0$ | 1 | DIRICHLET CONCENTRATION FOR $\boldsymbol{\pi}$ |
| $\boldsymbol{\lambda}_{\theta 1}, \boldsymbol{\lambda}_{\theta 2}$ | 1, 1 | BETA PRIOR ON $\theta_{kd}$ (BERNOULLI) |
| $\boldsymbol{\lambda}_{\gamma 1}, \boldsymbol{\lambda}_{\gamma 2}$ | 0.5, 0.1 | BETA PRIOR ON FEATURE WEIGHT $\gamma_{kd}$ |
| $\boldsymbol{\lambda}_w, \boldsymbol{\lambda}_{\bar{w}}$ | (1, 1) | GAMMA PRIOR ON CONSTRAINT WEIGHTS |
| $M$ | 500 | SVI MINI-BATCH SIZE |
| $\tau$ | 1 | SVI DELAY PARAMETER |
| $\kappa$ | 0.7 | SVI FORGETTING RATE |
| REL_TOL | $10^{-5}$ | CONVERGENCE TOLERANCE FOR $\Delta_{\text{EMA}}^{(t)}$ |
| WARMUP_PROP | 0.0 | PROPORTION OF ITERATIONS UNDER FIXED $w_{nm}=1$ |

## A.4. Evaluation Metrics

We report results using two widely adopted external metrics for clustering evaluation:

**Normalized Mutual Information (NMI)** measures the information-theoretic agreement between predicted and ground-truth partitions,

$$\text{NMI}(Y, Z) = \frac{2 \cdot I(Y; Z)}{H(Y) + H(Z)}, \tag{A.13}$$

where $I(Y; Z)$ is the mutual information between ground-truth labels $Y$ and cluster assignments $Z$, and $H(\cdot)$ denotes entropy.

**Clustering Accuracy (ACC)** evaluates the maximum matching between cluster assignments and class labels,

$$\text{ACC} = \max_{m \in \mathcal{M}} \frac{1}{N} \sum_{n=1}^{N} \mathbb{I}\{y_n = m(z_n)\}, \tag{A.14}$$

where $y_n$ denotes the ground-truth label, $z_n$ is the cluster assignment, and $\mathcal{M}$ is the set of all possible one-to-one mappings between cluster labels and classes (solved via the Hungarian algorithm).

## A.5. Outline of the Gaussian Extension

The BASIL framework can be extended to continuous-valued data with a Gaussian likelihood, requiring only localised modifications to the cluster-parameter and potential blocks while preserving the HMRF graph, the constraint-weight updates, the mixture-weight update, and the assignment update for $\Phi_{nk}$.

1. Replace the per-feature Bernoulli likelihood $\text{Bern}(x_{nd} \,|\, \tilde{\theta}_{kd})$ with a per-feature univariate Gaussian $\mathcal{N}(x_{nd} \,|\, \tilde{\mu}_{kd}, \tilde{\sigma}_{kd}^2)$, and replace the Beta prior on $\theta_{kd}$ with the standard Normal–Inverse-Gamma conjugate prior on $(\mu_{kd}, \sigma_{kd}^2)$, paralleling the per-feature Bernoulli–Beta structure and the mean-field independence across features.

2. The feature-relevance shrinkage of Section 4 carries over identically as $\tilde{\mu}_{kd} = \gamma_{kd}\mu_{kd} + (1 - \gamma_{kd})\mu_{0d}$, with the population-level mean $\mu_{0d}$ replacing $\theta_{0d}$.

3. The cluster-parameter update remains closed-form conjugate, with Normal–Inverse-Gamma posterior sufficient statistics weighted by $\mathbb{E}_q[\gamma_{kd}]$ in direct analogy to the Bernoulli–Beta derivation of Equation (C.23).

4. The feature-importance of Equation (10) applies unchanged, with the relevance score $\hat{l}_{kd}$ evaluated at the current Gaussian likelihood and Normal–Inverse-Gamma variational posterior.

5. The KL-divergence potential of Equation (7) admits a closed form for Gaussians, given per-feature by the symmetric KL between $\mathcal{N}(\tilde{\mu}_{kd}, \tilde{\sigma}_{kd}^2)$ and $\mathcal{N}(\tilde{\mu}_{jd}, \tilde{\sigma}_{jd}^2)$ summed across feature dimensions $d$.

Algorithm 1 and all remaining updates ($\boldsymbol{\pi}$, $w_{nm}$, $\bar{w}_{nm}$, $\Phi_{nk}$) are untouched. For mixed-type data, a product likelihood combining Bernoulli factors over binary features and Gaussian factors over continuous features yields independent conjugate updates per feature, with the feature-relevance shrinkage and constraint-weight machinery unchanged.

# B. Additional Results

## B.1. Simulations for Sanity Check

To validate our implementation and to test whether the HMRF Bayesian framework behaves well under semi-supervision before extending it to a Bernoulli likelihood, we first evaluated the SS-DPGMM baseline (Echraibi et al., 2019) on synthetic 2D Gaussian data (Figure B.1). As shown in Table B.3, clustering performance improves monotonically as the proportion of labelled constraints increases, with unlabelled-data NMI rising from $0.76$–$0.79$ unsupervised to $0.86$–$0.87$ at $50\%$ supervision across sample sizes $N \in \{500, 5,000, 50,000\}$. This indicates that the Bayesian semi-supervised mixture family is well-behaved under Gaussian likelihoods, and the main-body benchmarks then focus on the Bernoulli likelihood that suits our downstream binary applications.

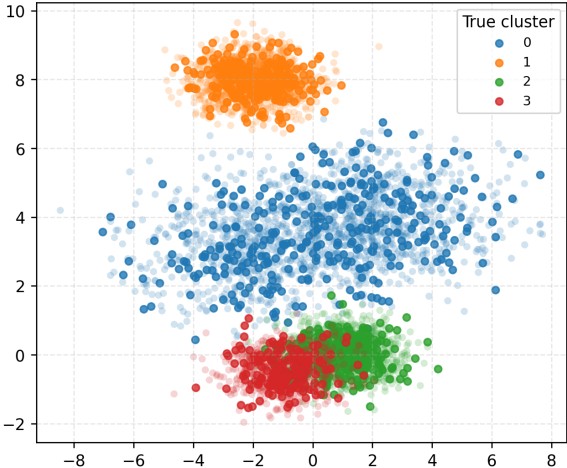

*Figure B.1.* Simulated 2D Gaussian data with 20% labeled.

Subsequently, we extended the framework to accommodate binary features using a Bernoulli likelihood. We evaluated the proposed BASIL model, which integrates both must-link and cannot-link constraints within an SVI framework, focusing on gains in clustering performance and computational efficiency. The parameters for the simulated Bernoulli distributions are provided in Table B.4.

Comparing the results in Table B.5 and Table B.6, we observe that while BASIL incurs a marginal performance penalty in some regimes, it achieves higher NMIs on larger datasets when only $20\%$ of samples are labelled, specifically by leveraging cannot-link information. Furthermore, the SVI-based mini-batch learning scheme enables the model to scale efficiently to large $N$, resulting in a substantial reduction in training overhead.

Finally, we conducted a sanity check for the integrated feature selection mechanism (data and results in Tables B.7 and B.9). The results indicate that the inclusion of feature selection maintains performance on larger data while yielding noticeable gains in scenarios with sparse supervision (e.g., $20\%$ labelled samples).

We further compare BASIL to baseline models (Table B.10). In comparison with the baselines on synthetic data, BASIL achieves higher test NMI scores (e.g., $0.77$–$0.81$ vs. $0.75$–$0.76$ for PCK/MPCK, an improvement of $0.02$–$0.05$) while offering a substantial computational speed-up over the baselines in large-scale settings. By leveraging mini-batch learning with a batch size of $M = 500$, our approach achieves an approximately $98\%$ reduction in training time compared to full-batch methods for large-scale datasets where $N \geq 50,000$ (e.g., $19.42$ minutes vs. $18.95$ hours for full-batch), with marginal degradation in test NMI ($0.01$–$0.03$). For the MML baseline running on large-scale datasets, we implement a bootstrap resampling strategy with 10 iterations to estimate the average distance metrics from subsets of 5,000 samples (1,000 for MNIST). The resulting metrics are subsequently utilized for Kmeans clustering on the full large-scale datasets where $N \geq 50,000$. This subsampling procedure is critical for mitigating memory constraints and ensuring computational tractability when processing high-volume data regimes.

*Table B.3.* NMIs and Time of SS-DPGMM Aggregated over 10 Runs per GMM Dataset Initialized by Kmeans.

| LABEL% | 0% | | | 20% | | | 50% | | |
|---|---|---|---|---|---|---|---|---|---|
| SIZE | LABEL | UNLABEL | TIME | LABEL | UNLABEL | TIME | LABEL | UNLABEL | TIME |
| 500 | – | 0.76± 0.00 | 5.51 s± 0.97 | 0.91± 0.00 | 0.84± 0.00 | 2.09 s± 0.01 | 0.89± 0.00 | 0.86± 0.00 | 3.87 s± 0.04 |
| 5,000 | – | 0.79± 0.00 | 4.13 m± 0.24 | 0.89± 0.00 | 0.87± 0.00 | 2.55 m± 0.54 | 0.89± 0.00 | 0.87± 0.00 | 6.16 m± 0.43 |
| 50,000 | – | 0.78± 0.00 | 1.11 h± 0.01 | 0.89± 0.00 | 0.86± 0.00 | 5.11 h± 0.16 | 0.89± 0.00 | 0.86± 0.00 | 11.92 h± 0.11 |

*Table B.4.* The Mean of Synthetic Binary Data.

| CLUSTER | X1 | X2 | X3 | X4 | X5 | X6 | LABEL |
|---|---|---|---|---|---|---|---|
| A | 0.90 | 0.95 | 0.90 | 0.01 | 0.01 | 0.01 | 0 |
| B | 0.35 | 0.35 | 0.35 | 0.01 | 0.01 | 0.01 | 0 |
| C | 0.01 | 0.01 | 0.01 | 0.95 | 0.95 | 0.95 | 1 |
| D | 0.90 | 0.90 | 0.90 | 0.90 | 0.95 | 0.95 | 2 |
| E | 0.90 | 0.90 | 0.90 | 0.75 | 0.70 | 0.70 | 3 |

*Table B.5.* NMIs and Time of SS-DPGMM Aggregated over 10 Runs per BMM Dataset Initialized by Kmedoids with Hamming Distance.

| LABEL% | 0% | | | 20% | | | 50% | | |
|---|---|---|---|---|---|---|---|---|---|
| SIZE | LABEL | UNLABEL | TIME | LABEL | UNLABEL | TIME | LABEL | UNLABEL | TIME |
| 500 | – | 0.71± 0.00 | 1.19 s± 0.02 | 0.87± 0.00 | 0.79± 0.00 | 3.99 s± 0.06 | 0.89± 0.00 | 0.80± 0.00 | 3.95 s± 0.04 |
| 5,000 | – | 0.73± 0.00 | 0.53 m± 0.00 | 0.89± 0.00 | 0.74± 0.00 | 3.87 m± 0.30 | 0.89± 0.00 | 0.82± 0.00 | 8.81 m± 0.68 |
| 50,000 | – | 0.71± 0.00 | 8.12 m± 1.78 | 0.89± 0.00 | 0.73± 0.00 | 9.10 h± 0.24 | 0.89± 0.00 | 0.80± 0.00 | 18.95 h± 0.26 |

*Table B.6.* NMIs and Time of BASIL Aggregated over 10 Runs per BMM Dataset Initialized by Kmedoids with Hamming Distance (Large Data Using Kmodes) with Batch Size 500.

| LABEL% | 0% | | | 20% | | | 50% | | |
|---|---|---|---|---|---|---|---|---|---|
| SIZE | LABEL | UNLABEL | TIME | LABEL | UNLABEL | TIME | LABEL | UNLABEL | TIME |
| 500 | – | 0.70± 0.00 | 0.44 s± 0.01 | 0.85± 0.00 | 0.78± 0.00 | 8.61 s± 0.10 | 0.87± 0.00 | 0.77± 0.00 | 78.11 s± 0.97 |
| 5,000 | – | 0.72± 0.00 | 1.08 s± 0.07 | 0.96± 0.01 | 0.79± 0.01 | 2.25 m± 0.13 | 0.94± 0.03 | 0.77± 0.00 | 13.74 m± 0.28 |
| 50,000 | – | 0.71± 0.00 | 3.98 s± 0.11 | 0.98± 0.01 | 0.78± 0.01 | 13.84 m± 0.96 | 0.99± 0.00 | 0.77± 0.00 | 19.42 m± 1.32 |
| 500,000 | – | 0.71± 0.00 | 1.59 m± 0.58 | 0.98± 0.00 | 0.78± 0.01 | 24.25 m± 1.89 | 0.93± 0.00 | 0.77± 0.00 | 24.83 m± 1.84 |

*Table B.7.* The Mean of Synthetic Binary Data for Feature Importance Testing.

| CLUSTER | X1 | X2 | X3 | X4 | X5 | X6 | X7 | X8 | X9 | X10 | X11 | X12 | LABEL |
|---|---|---|---|---|---|---|---|---|---|---|---|---|---|
| | INFORMATIVE | | | IRRELEVANT | | | | | NONINFORMATIVE | | | | |
| A | 0.9 | 0.9 | 0.9 | 0.8 | 0.8 | 0.2 | 0.8 | 0.2 | 0.5 | 0.5 | 0.5 | 0.5 | 0 |
| B | 0.9 | 0.9 | 0.9 | 0.2 | 0.2 | 0.8 | 0.2 | 0.8 | 0.5 | 0.5 | 0.5 | 0.5 | 0 |
| C | 0.2 | 0.2 | 0.2 | 0.8 | 0.8 | 0.2 | 0.8 | 0.2 | 0.5 | 0.5 | 0.5 | 0.5 | 1 |
| D | 0.2 | 0.2 | 0.2 | 0.2 | 0.2 | 0.8 | 0.2 | 0.8 | 0.5 | 0.5 | 0.5 | 0.5 | 1 |

*Table B.8.* Quantitative feature relevance on the synthetic FS benchmark. Mean recovered $\max_k \mathbb{E}_q[\gamma_{kd}]$ within each ground-truth feature group, and the AUROC for separating informative from noninformative features. Aggregated over 10 random seeds at 20% supervision.

| SETTING | INFORMATIVE | IRRELEVANT | NONINFORMATIVE | AUROC |
|---|---|---|---|---|
| $N = 500$ | 0.971 | 0.171 | 0.043 | 1.00 |
| $N = 5,000$ | 0.975 | 0.005 | 0.005 | 1.00 |
| $N = 50,000$ | 0.997 | 0.663 | 0.020 | 1.00 |

*Table B.9.* NMIs and Time of Models Aggregated over 10 Runs per BMM Dataset for Feature Selection Initialized by Kmedoids with Hamming Distance with Batch Size 500.

| | | | LABEL% 0% | | 20% | | 50% | |
|---|---|---|---|---|---|---|---|---|
| SIZE | MODEL | FEATSELECT | TRAIN | TEST | TRAIN | TEST | TRAIN | TEST |
| 500 | MPCK | | – | – | $0.72\pm 0.00$ | $0.69\pm 0.00$ | $0.89\pm 0.00$ | $0.69\pm 0.00$ |
| | BASIL | NO | $0.02\pm 0.00$ | $0.02\pm 0.00$ | $0.61\pm 0.00$ | $0.64\pm 0.00$ | $0.83\pm 0.00$ | $0.68\pm 0.00$ |
| | BASIL | YES | $0.02\pm 0.00$ | $0.02\pm 0.00$ | $0.72\pm 0.00$ | $0.69\pm 0.00$ | $0.89\pm 0.00$ | $0.69\pm 0.00$ |
| 5,000 | BASIL | NO | $0.01\pm 0.00$ | $0.01\pm 0.00$ | $0.70\pm 0.01$ | $0.69\pm 0.01$ | $0.80\pm 0.00$ | $0.69\pm 0.00$ |
| | BASIL | YES | $0.01\pm 0.00$ | $0.01\pm 0.00$ | $0.74\pm 0.00$ | $0.69\pm 0.00$ | $0.80\pm 0.00$ | $0.69\pm 0.00$ |
| 50,000 | BASIL | NO | $0.00\pm 0.00$ | $0.01\pm 0.00$ | $0.70\pm 0.01$ | $0.69\pm 0.01$ | $0.80\pm 0.00$ | $0.69\pm 0.00$ |
| | BASIL | YES | $0.00\pm 0.00$ | $0.01\pm 0.00$ | $0.71\pm 0.00$ | $0.69\pm 0.00$ | $0.80\pm 0.00$ | $0.69\pm 0.00$ |

*Table B.10.* NMIs and Time of Models Aggregated over 10 Runs per BMM Dataset and the Same Test Dataset.

| | | LABEL% 20% | | | 50% | | |
|---|---|---|---|---|---|---|---|
| SIZE | MODEL | TRAIN | TEST | TIME | TRAIN | TEST | TIME |
| 500 | MML | $0.60\pm 0.05$ | $0.60\pm 0.06$ | $0.11\,\mathrm{s}\pm 0.06$ | $0.57\pm 0.00$ | $0.56\pm 0.00$ | $0.17\,\mathrm{s}\pm 0.01$ |
| | PCK | $0.77\pm 0.01$ | $0.76\pm 0.00$ | $0.81\,\mathrm{s}\pm 0.01$ | $0.83\pm 0.01$ | $0.76\pm 0.00$ | $30.02\,\mathrm{s}\pm 0.45$ |
| | MPCK | $0.77\pm 0.01$ | $0.76\pm 0.00$ | $4.38\,\mathrm{s}\pm 1.75$ | $0.81\pm 0.03$ | $0.75\pm 0.02$ | $24.42\,\mathrm{s}\pm 2.60$ |
| | BASIL | $0.75\pm 0.00$ | $\mathbf{0.80}\pm 0.00$ | $8.62\,\mathrm{s}\pm 0.10$ | $0.81\pm 0.00$ | $\mathbf{0.78}\pm 0.00$ | $78.12\,\mathrm{s}\pm 0.97$ |
| 5,000 | MML | $0.60\pm 0.06$ | $0.62\pm 0.07$ | $1.51\,\mathrm{s}\pm 0.07$ | $0.57\pm 0.05$ | $0.58\pm 0.06$ | $13.86\,\mathrm{s}\pm 0.06$ |
| | PCK | $0.77\pm 0.02$ | $0.76\pm 0.00$ | $2.54\,\mathrm{h}\pm 0.14$ | – | – | $>2\,\mathrm{d}$ |
| | MPCK | $0.78\pm 0.01$ | $0.76\pm 0.00$ | $2.42\,\mathrm{h}\pm 0.08$ | – | – | $>2\,\mathrm{d}$ |
| | BASIL | $0.80\pm 0.01$ | $\mathbf{0.81}\pm 0.00$ | $2.25\,\mathrm{m}\pm 0.13$ | $0.82\pm 0.01$ | $\mathbf{0.78}\pm 0.00$ | $13.74\,\mathrm{m}\pm 0.28$ |
| 50,000 | MML | $0.55\pm 0.07$ | $0.60\pm 0.07$ | $1.18\,\mathrm{m}\pm 0.02$ | $0.51\pm 0.11$ | $0.53\pm 0.11$ | $1.18\,\mathrm{m}\pm 0.02$ |
| | PCK | – | – | $>2\,\mathrm{d}$ | – | – | $>2\,\mathrm{d}$ |
| | MPCK | – | – | $>2\,\mathrm{d}$ | – | – | $>2\,\mathrm{d}$ |
| | BASIL | $0.78\pm 0.01$ | $\mathbf{0.81}\pm 0.01$ | $13.84\,\mathrm{m}\pm 0.96$ | $0.85\pm 0.00$ | $\mathbf{0.77}\pm 0.00$ | $19.42\,\mathrm{m}\pm 1.32$ |
| 500,000 | MML | $0.50\pm 0.08$ | $0.51\pm 0.10$ | $1.13\,\mathrm{m}\pm 0.01$ | $0.49\pm 0.16$ | $0.50\pm 0.17$ | $1.13\,\mathrm{m}\pm 0.01$ |
| | PCK | – | – | $>2\,\mathrm{d}$ | – | – | $>2\,\mathrm{d}$ |
| | MPCK | – | – | $>2\,\mathrm{d}$ | – | – | $>2\,\mathrm{d}$ |
| | BASIL | $0.80\pm 0.01$ | $\mathbf{0.79}\pm 0.01$ | $24.25\,\mathrm{m}\pm 1.89$ | $0.83\pm 0.00$ | $\mathbf{0.77}\pm 0.00$ | $24.83\,\mathrm{m}\pm 1.84$ |

## B.2. Quantitative Interpretability Details

This appendix expands the AUROC analyses inlined in the Interpretability and Adaptive Constraint Weighting paragraphs of Section 6.1 with the supporting per-noise and per-stratum tables.

**Feature relevance.** Using the synthetic FS benchmark of Table B.7, we trained BASIL with feature selection enabled for 10 random seeds at each sample size $N \in \{500, 5{,}000, 50{,}000\}$ and 20% supervision. For every recovered cluster $k$ we treat $\mathbb{E}_q[\gamma_{kd}]$ as a per-feature score and report the mean recovered relevance within each ground-truth feature group together with the AUROC for separating informative from noninformative features (Table B.8). The recovered $\mathbb{E}_q[\gamma_{kd}]$ values cluster tightly near 1.0 for the informative block ($X_1$–$X_3$) and near 0.0 for the noninformative block ($X_9$–$X_{12}$), with intermediate values for the cluster-discriminative-but-not-class-discriminative features ($X_4$–$X_8$). Across all sample sizes the AUROC is 1.00.

**Constraint reliability.** For the corrupted-DIGITS experiment of Section 6.1 at 20% supervision, we trained BASIL with adaptive constraint weights $w_{nm} \sim \mathrm{Gamma}(1, 2.5)$ and feature selection for 10 seeds across noise levels $0\%, 10\%, 20\%, 30\%, 50\%$. The posterior mean is rescaled by the prior mean as $\tilde{\mathbb{E}}_q[w_{nm}] = (\beta_{nm,1}/\beta_{nm,2})/\mathbb{E}[\mathrm{Gamma}(1, 2.5)]$ to lie in $[0, 1]$, where 1 indicates full prior confidence. Table B.11 reports the test ACC and the constraint-reliability AUROC as a function of noise level. Clean constraints retain $\tilde{\mathbb{E}}_q[w] \approx 1.0$ while corrupted constraints are progressively down-weighted, and the AUROC degrades gracefully from 0.97 at 10% noise to 0.91 at 50% noise. At 20% noise the overall AUROC further decomposes into must-link $0.93 \pm 0.01$ and cannot-link $0.97 \pm 0.01$, indicating that the mechanism resolves cannot-link violations more reliably than must-link violations under noise.

*Table B.11.* Posterior constraint-reliability AUROC across noise levels on DIGITS at 20% supervision under $w_{nm} \sim \text{Gamma}(1, 2.5)$. Aggregated over 10 random seeds. $\tilde{\mathbb{E}}_q[w]$ is the rescaled posterior weight.

| NOISE | TEST ACC | $\tilde{\mathbb{E}}_q[w]$ CLEAN | $\tilde{\mathbb{E}}_q[w]$ CORRUPT | AUROC |
|---|---|---|---|---|
| 0% | $0.80\pm 0.03$ | 0.999 | – | – |
| 10% | $0.77\pm 0.04$ | 0.997 | 0.914 | $0.969\pm 0.012$ |
| 20% | $0.76\pm 0.02$ | 0.997 | 0.872 | $0.953\pm 0.017$ |
| 30% | $0.76\pm 0.03$ | 0.996 | 0.852 | $0.947\pm 0.010$ |
| 50% | $0.71\pm 0.04$ | 0.992 | 0.858 | $0.914\pm 0.016$ |

**Hard-versus-easy preservation.**    A natural concern is whether adaptive weighting penalises difficult-to-cluster constraints rather than genuinely noisy ones. Partitioning constraints by the Euclidean distance between their endpoints in feature space (top vs bottom quartile) and by their corruption status, Table B.12 reports rescaled $\tilde{\mathbb{E}}_q[w_{nm}]$ stratified by both axes on DIGITS at 20% supervision. Within each difficulty class, clean constraints consistently receive higher weight than corrupted ones (a $\sim 12\text{–}20$ point gap among hard constraints), confirming that the mechanism detects violation rather than difficulty per se. The blind spot is the easy-corrupt cell, where corrupted constraints sitting near coherent clusters become indistinguishable from clean ones.

*Table B.12.* Rescaled $\tilde{\mathbb{E}}_q[w_{nm}]$ stratified by constraint difficulty (top vs bottom quartile of feature-space distance) and corruption status on DIGITS at 20% supervision. Within each difficulty class, clean constraints receive higher weight than corrupted ones. Aggregated over 10 random seeds.

| NOISE | HARD CLEAN | HARD CORRUPT | EASY CLEAN | EASY CORRUPT | AUROC |
|---|---|---|---|---|---|
| 20% | 0.987 | 0.785 | 1.000 | 0.974 | 0.963 |
| 30% | 0.909 | 0.791 | 0.989 | 0.989 | 0.808 |

## B.3. Open Digit Datasets for Ablation Study

We performed an ablation study to systematically evaluate the impact of two KL-divergence penalties, the learning of constraint weights ($w_{nm}$), and the inclusion of feature selection (FS). Table B.13 summarizes the Normalized Mutual Information (NMI) scores and clustering accuracy (ACC) for each configuration. We observed that the configuration incorporating feature selection plus a KL-divergence penalty on the posterior data distribution $p(\mathbf{x}_n)$ (denoted as D), achieved better performance across both metrics. Conversely, applying the KL-divergence between the variational distributions of component parameters $q(\boldsymbol{\theta}_k)$ (KL) has reduced performance. Moreover, simultaneously learning the constraint weights $w_{nm}$ can also cause the majority of weights to vanish ($w_{nm} \to 0$), thereby diminishing the utility of supervised signals. Notably, fixing the constraint weights ($w_{nm} = 1$) instead of learning them significantly enhanced model stability, as evidenced by the reduced standard errors (SE). Furthermore, the integrated feature selection mechanism provides two key advantages: it identifies highly discriminative features to enhance interpretability and effectively accelerates the convergence rate of the inference process. In summary, feature selection emerges as the most critical component, whereas the original KL-divergence formulation proposed by Echraibi et al. (2019) exhibits incompatibility with constraint weight learning. Consequently, we adopt the modified KL-divergence formulation in BASIL. We further evaluate model performance on the MNIST dataset (Table B.21), where fixed constraint weights yield higher results under reliable supervision. Moreover, jointly incorporating feature selection with KL-divergence distance learning further enhances clustering accuracy, particularly at higher supervision ratios.

**When adaptive weighting helps and when it hurts.**    In clean regimes ($< 1\%$ noise), fixing $w_{nm}=1$ matches or slightly outperforms adaptive weighting (Table B.13), since the additional flexibility introduces optimisation variability without signal to exploit. As the corrupted fraction rises, adaptive weighting becomes increasingly valuable. At 30% noise it recovers $\sim 8$ accuracy points over a fixed-weight baseline (Figure 3), while at 50% noise (an essentially random label setting) both modes collapse toward unsupervised performance. Adaptive weighting is also less effective on very large $N$ when the constraint budget is sparse relative to the data (e.g. MNIST with $\sim 5\times10^6$ pairs out of $\sim 7.2\times10^7$, $\approx 7\%$), because each sample sees too few constraint neighbours per batch for the Gamma posterior over $w_{nm}$ to concentrate.

**Hyperparameter sensitivity of $\lambda_{w2}$.** The rate parameter $\lambda_{w2}$ acts as a single interpretable knob. Values $\lambda_{w2} \in [1.5, 5]$ cover most regimes (Tables B.15 and B.16), with $\lambda_{w2} \approx 1.5$ appropriate for $< 1\%$ noise and $\lambda_{w2} \approx 2.5$ for $\geq 20\%$ noise. The default $\mathrm{Gamma}(1,1)$ is appropriate when the noise level is unknown. The practical recipe is therefore to use fixed weights when constraints are trusted and adaptive weights with a moderately informative prior when reliability is uncertain.

*Table B.13.* NMI, Clustering Accuracy and Time for Ablation Study of BASIL Model Aggregated over 10 Runs for DIGITS Dataset and the Same Test Dataset.

| | LABEL% | 0% | | | 20% | | | 50% | | |
|---|---|---|---|---|---|---|---|---|---|---|
| METRIC | MODEL | TRAIN | TEST | TIME | TRAIN | TEST | TIME | TRAIN | TEST | TIME |
| NMI | $(w_{nm},$KL$)$ | 0.69±0.00 | 0.72±0.00 | 21.13 s±0.31 | 0.54±0.03 | 0.71±0.03 | 23.95 m±1.77 | 0.46±0.06 | 0.67±0.03 | 1.73 h±0.07 |
| | $(w_{nm},$FS$)$** | 0.69±0.02 | 0.72±0.01 | 5.81 s±0.34 | 0.74±0.02 | **0.77**±0.01 | 6.63 m±0.54 | 0.86±0.00 | 0.79±0.01 | 0.74 h±0.03 |
| | $(w_{nm},$KL+FS$)$ | 0.68±0.01 | 0.72±0.01 | 9.00 s±0.57 | 0.61±0.04 | 0.66±0.04 | 31.37 m±0.32 | 0.29±0.05 | 0.52±0.04 | 1.73 h±0.07 |
| | $(w_{nm},$D$)$ | 0.69±0.00 | 0.72±0.00 | 21.13 s±0.31 | 0.73±0.01 | **0.77**±0.01 | 15.62 m±0.19 | 0.78±0.04 | 0.78±0.02 | 1.64 h±0.02 |
| | $(w_{nm},$D+FS$)$* | 0.68±0.01 | 0.72±0.01 | 9.00 s±0.57 | 0.72±0.01 | **0.77**±0.02 | 6.68 m±0.60 | 0.78±0.04 | 0.78±0.02 | 0.77 h±0.04 |
| | $(1,$KL$)$ | 0.69±0.00 | 0.72±0.00 | 11.19 s±0.15 | 0.69±0.03 | 0.73±0.03 | 15.60 m±0.20 | 0.86±0.00 | 0.79±0.00 | 1.23 h±0.02 |
| | $(1,$FS$)$** | 0.69±0.01 | 0.72±0.01 | 5.85 s±0.32 | 0.75±0.02 | 0.76±0.02 | 7.04 m±0.25 | 0.86±0.00 | **0.80**±0.00 | 0.74 h±0.02 |
| | $(1,$KL+FS$)$ | 0.69±0.01 | 0.72±0.01 | 5.90 s±0.29 | 0.77±0.00 | 0.73±0.00 | 6.99 m±0.34 | 0.86±0.00 | 0.78±0.01 | 0.75 h±0.02 |
| | $(1,$D$)$** | 0.69±0.00 | 0.72±0.00 | 11.19 s±0.15 | 0.75±0.01 | **0.77**±0.01 | 15.49 m±0.14 | 0.86±0.00 | **0.80**±0.00 | 1.63 h±0.02 |
| | $(1,$D+FS$)$** | 0.69±0.01 | 0.72±0.01 | 5.90 s±0.29 | 0.75±0.01 | **0.77**±0.01 | 7.00 m±0.37 | 0.86±0.00 | **0.80**±0.00 | 0.76 h±0.02 |
| | $(1,-)$ | 0.69±0.01 | 0.72±0.01 | 5.90 s±0.29 | 0.75±0.00 | 0.75±0.01 | 8.42 m±0.05 | 0.87±0.00 | 0.79±0.00 | 0.89 h±0.01 |
| ACC | $(w_{nm},$KL$)$ | 0.68±0.00 | 0.68±0.00 | 21.13 s±0.31 | 0.61±0.04 | 0.68±0.04 | 23.95 m±1.77 | 0.55±0.04 | 0.69±0.04 | 1.73 h±0.07 |
| | $(w_{nm},$FS$)$* | 0.69±0.02 | 0.68±0.01 | 5.81 s±0.34 | 0.79±0.05 | 0.78±0.04 | 6.63 m±0.54 | 0.93±0.00 | 0.86±0.00 | 0.74 h±0.03 |
| | $(w_{nm},$KL+FS$)$ | 0.69±0.02 | 0.68±0.01 | 9.00 s±0.57 | 0.67±0.05 | 0.64±0.05 | 31.37 m±0.32 | 0.38±0.04 | 0.47±0.04 | 1.73 h±0.07 |
| | $(w_{nm},$D$)$ | 0.68±0.00 | 0.68±0.00 | 21.13 s±0.31 | 0.76±0.04 | 0.75±0.03 | 15.62 m±0.19 | 0.84±0.05 | 0.82±0.04 | 1.64 h±0.02 |
| | $(w_{nm},$D+FS$)$ | 0.69±0.02 | 0.68±0.01 | 9.00 s±0.57 | 0.77±0.02 | 0.76±0.03 | 6.68 m±0.60 | 0.84±0.06 | 0.82±0.04 | 0.77 h±0.04 |
| | $(1,$KL$)$ | 0.68±0.00 | 0.68±0.00 | 11.19 s±0.15 | 0.74±0.04 | 0.75±0.04 | 15.60 m±0.20 | 0.93±0.00 | 0.86±0.00 | 1.23 h±0.02 |
| | $(1,$FS$)$** | 0.69±0.02 | 0.68±0.01 | 5.85 s±0.32 | 0.83±0.03 | **0.80**±0.02 | 7.04 m±0.25 | 0.93±0.00 | **0.87**±0.00 | 0.74 h±0.02 |
| | $(1,$KL+FS$)$ | 0.69±0.02 | 0.68±0.01 | 5.90 s±0.29 | 0.87±0.00 | 0.76±0.00 | 6.99 m±0.34 | 0.93±0.00 | 0.85±0.01 | 0.75 h±0.02 |
| | $(1,$D$)$** | 0.68±0.00 | 0.68±0.00 | 11.19 s±0.15 | 0.82±0.01 | **0.80**±0.03 | 15.49 m±0.14 | 0.93±0.00 | **0.87**±0.00 | 1.63 h±0.02 |
| | $(1,$D+FS$)$** | 0.69±0.02 | 0.68±0.01 | 5.90 s±0.29 | 0.83±0.02 | **0.80**±0.02 | 7.00 m±0.37 | 0.93±0.00 | **0.87**±0.00 | 0.76 h±0.02 |
| | $(1,-)$** | 0.69±0.02 | 0.68±0.01 | 5.90 s±0.29 | 0.82±0.00 | **0.80**±0.00 | 8.42 m±0.05 | 0.94±0.00 | **0.87**±0.00 | 0.89 h±0.01 |

To be more specific, experimental evaluations on the MNIST dataset imply that the inclusion of the KL-divergence penalty on the data distribution (D) achieves higher performance in high-supervision regimes (e.g., $50\%$ labelling). Notably, this configuration eliminates the necessity for extensive tuning of the feature weight prior $\gamma_{kd}$ and constraint weights, especially for large-scale datasets. In contrast, the standalone feature selection (FS) variant exhibits significant sensitivity to prior specification. Specifically, the FS-only model mischaracterizes the discriminative feature manifold for digit "9", failing to recover the correct feature importance heatmap as shown in Figure B.2. This indicates that distance measure on cluster divergence acts as a "soft supervision signal" that guides the clustering algorithm to learn cluster assignments and the feature relevance that best fits constraints, by adapting the penalty based on cluster divergence (Figure B.3).

To evaluate the robustness of our approach under unreliable supervision, we conduct an additional ablation study using a version of the DIGITS dataset corrupted with different levels of noisy constraints (Tables B.14 to B.16). We further evaluate model performance on the corrupted version of MNIST (Table B.22). The empirical results on the DIGITS dataset again demonstrate that the feature selection (FS) and its variants (D+FS) achieve the highest performance and stability in low-noise regimes ($< 1\%$) when bypassing the estimation of constraint weights ($w_{nm} = 1$). However, as the proportion of erroneous constraints increases to $50\%$, model performance degrades substantially, converging toward unsupervised clustering behaviour, except when prior hyperparameters are carefully tuned. Across all noise regimes, the adaptive estimation of latent constraint weights ($w_{nm}$) combined with feature selection proves beneficial, particularly when the weight prior hyperparameters are properly calibrated, enabling the model to distinguish between reliable and misleading constraints.

However, combining all three components can lead to over-regularisation in low-noise regimes ($0\%$ to $30\%$) with limited supervision ($20\%$), indicating a trade-off between model complexity and regularisation strength. This arises from a conflict between constraint weighting and distance learning: constraint weighting tends to down-weight violated constraints to mitigate their effects on clustering, whereas distance learning seeks to transform to a feature space that best satisfies the constraints. In certain cases, constraint weighting may incorrectly down-weight a valid constraint, while distance learning

subsequently adapts to this erroneous configuration, yielding suboptimal results. In the presence of increasing proportions of unreliable constraints, we observe that a phased optimization strategy combining fixed-weight regularization (1+D+FS) with subsequent adaptive weight learning ($w_{nm}$+FS) yields the most robust results. This approach utilizes an initial warmup phase wherein the model employs fixed unit weights and a KL divergence penalty on the data distribution to stabilize the feature relevance manifold. Following this stabilization, the model transitions to the active inference of latent constraint weights ($w_{nm}$), effectively allowing the framework to isolate and down-weight noisy supervisory signals during the later stages of convergence. Empirical performance under this scheme is sensitive to the warmup duration, requiring careful calibration of the phase transition proportion to balance feature selection stability with noise resilience.

In summary, when there is high confidence in the accuracy of the provided constraints, the optimal model configuration integrates simultaneous feature selection with a KL divergence penalty on the posterior data distribution. In this high-fidelity regime, employing fixed unit constraint weights ensures maximum stability and performance, as it effectively leverages the supervisory signal while bypassing the optimization complexities and instability risks inherent in adaptive estimation frameworks. Conversely, when constraints are susceptible to noise, the adaptive estimation of latent weights offers essential flexibility by allowing the model to isolate and down-weight unreliable signals. To harness the strengths of both paradigms, these approaches can be integrated into a phased optimization strategy governed by a warmup proportion. This strategy stabilizes the feature relevance manifold during the initial stages before transitioning to noise-robust weight refinement for large-scale datasets with a high level of supervision.

We also performed baselines on the benchmark DIGITS data and its noisy-constraint versions to evaluate the robustness of those models (Tables B.17 to B.20). The results are summarised in the main body.

*Table B.14.* NMI, Clustering Accuracy and Time for Incorrect Constraints Test of BASIL Model Aggregated over 10 Runs for DIGITS Dataset ($< 1\%$ Wrong Labels) and the Same Test Dataset.

| | LABEL% | 20% | | | 50% | | |
|---|---|---|---|---|---|---|---|
| METRIC | MODEL | TRAIN | TEST | TIME | TRAIN | TEST | TIME |
| NMI | $(w_{nm}$,FS) | $0.75\pm0.01$ | $0.75\pm0.04$ | $6.62\,\text{m}\pm0.62$ | $0.86\pm0.00$ | $\mathbf{0.78}\pm0.00$ | $0.74\,\text{h}\pm0.02$ |
| | Gamma$(1,1.5)$** | $0.76\pm0.01$ | $\mathbf{0.78}\pm0.02$ | $6.51\,\text{m}\pm0.44$ | $0.86\pm0.01$ | $\mathbf{0.78}\pm0.01$ | $0.73\,\text{h}\pm0.03$ |
| | $(w_{nm}$,D+FS) | $0.74\pm0.02$ | $0.74\pm0.03$ | $6.73\,\text{m}\pm0.50$ | $0.76\pm0.04$ | $0.76\pm0.02$ | $0.77\,\text{h}\pm0.04$ |
| | $(1$,FS)* | $0.76\pm0.01$ | $0.77\pm0.02$ | $7.01\,\text{m}\pm0.23$ | $0.86\pm0.00$ | $\mathbf{0.78}\pm0.00$ | $0.73\,\text{h}\pm0.02$ |
| | $(1$,D+FS)* | $0.77\pm0.02$ | $0.76\pm0.02$ | $7.01\,\text{m}\pm0.31$ | $0.86\pm0.00$ | $\mathbf{0.78}\pm0.00$ | $0.75\,\text{h}\pm0.03$ |
| ACC | $(w_{nm}$,FS) | $0.80\pm0.03$ | $0.77\pm0.04$ | $6.60\,\text{m}\pm0.62$ | $0.93\pm0.00$ | $\mathbf{0.84}\pm0.00$ | $0.74\,\text{h}\pm0.02$ |
| | Gamma$(1,1.5)$** | $0.82\pm0.03$ | $0.79\pm0.01$ | $6.51\,\text{m}\pm0.44$ | $0.92\pm0.03$ | $\mathbf{0.84}\pm0.02$ | $0.73\,\text{h}\pm0.03$ |
| | $(w_{nm}$,D+FS) | $0.79\pm0.04$ | $0.75\pm0.04$ | $6.73\,\text{m}\pm0.50$ | $0.81\pm0.05$ | $0.79\pm0.04$ | $0.77\,\text{h}\pm0.04$ |
| | $(1$,FS)** | $0.84\pm0.03$ | $\mathbf{0.80}\pm0.01$ | $7.01\,\text{m}\pm0.23$ | $0.93\pm0.00$ | $\mathbf{0.84}\pm0.00$ | $0.73\,\text{h}\pm0.02$ |
| | $(1$,D+FS)** | $0.85\pm0.04$ | $0.79\pm0.01$ | $7.01\,\text{m}\pm0.31$ | $0.93\pm0.00$ | $\mathbf{0.84}\pm0.00$ | $0.75\,\text{h}\pm0.03$ |

*Table B.15.* NMI, Clustering Accuracy and Time for Incorrect Constraints Test of BASIL Model Aggregated over 10 Runs for DIGITS Dataset ($20\%$ Wrong Labels) and the Same Test Dataset.

| | LABEL% | 20% | | | 50% | | |
|---|---|---|---|---|---|---|---|
| METRIC | MODEL | TRAIN | TEST | TIME | TRAIN | TEST | TIME |
| NMI | $(w_{nm}$,FS) | $0.72\pm0.02$ | $0.73\pm0.01$ | $6.99\,\text{m}\pm0.40$ | $0.81\pm0.00$ | $0.76\pm0.00$ | $0.76\,\text{h}\pm0.05$ |
| | Gamma$(1,2.5(5))$** | $0.70\pm0.01$ | $\mathbf{0.76}\pm0.02$ | $6.75\,\text{m}\pm0.36$ | $0.79\pm0.00$ | $\mathbf{0.77}\pm0.00$ | $0.75\,\text{h}\pm0.06$ |
| | $(w_{nm}$,D+FS) | $0.71\pm0.02$ | $0.73\pm0.03$ | $6.85\,\text{m}\pm0.22$ | $0.70\pm0.05$ | $0.72\pm0.03$ | $0.72\,\text{h}\pm0.04$ |
| | $(1$,FS)* | $0.73\pm0.02$ | $0.74\pm0.02$ | $7.30\,\text{m}\pm0.44$ | $0.81\pm0.00$ | $0.76\pm0.00$ | $0.39\,\text{h}\pm0.02$ |
| | $(1$,D+FS)* | $0.73\pm0.01$ | $0.74\pm0.01$ | $7.17\,\text{m}\pm0.22$ | $0.81\pm0.00$ | $0.75\pm0.00$ | $0.39\,\text{h}\pm0.02$ |
| ACC | $(w_{nm}$,FS) | $0.76\pm0.05$ | $0.74\pm0.03$ | $6.99\,\text{m}\pm0.40$ | $0.90\pm0.00$ | $0.81\pm0.00$ | $0.76\,\text{h}\pm0.05$ |
| | Gamma$(1,2.5(5))$** | $0.80\pm0.02$ | $\mathbf{0.80}\pm0.03$ | $6.75\,\text{m}\pm0.36$ | $0.88\pm0.00$ | $\mathbf{0.82}\pm0.00$ | $0.75\,\text{h}\pm0.06$ |
| | $(w_{nm}$,D+FS) | $0.76\pm0.06$ | $0.73\pm0.04$ | $6.85\,\text{m}\pm0.22$ | $0.77\pm0.06$ | $0.74\pm0.05$ | $0.72\,\text{h}\pm0.04$ |
| | $(1$,FS)* | $0.80\pm0.04$ | $0.76\pm0.04$ | $7.30\,\text{m}\pm0.44$ | $0.90\pm0.00$ | $0.81\pm0.00$ | $0.39\,\text{h}\pm0.02$ |
| | $(1$,D+FS)* | $0.80\pm0.04$ | $0.76\pm0.02$ | $7.17\,\text{m}\pm0.22$ | $0.90\pm0.00$ | $0.81\pm0.00$ | $0.39\,\text{h}\pm0.02$ |

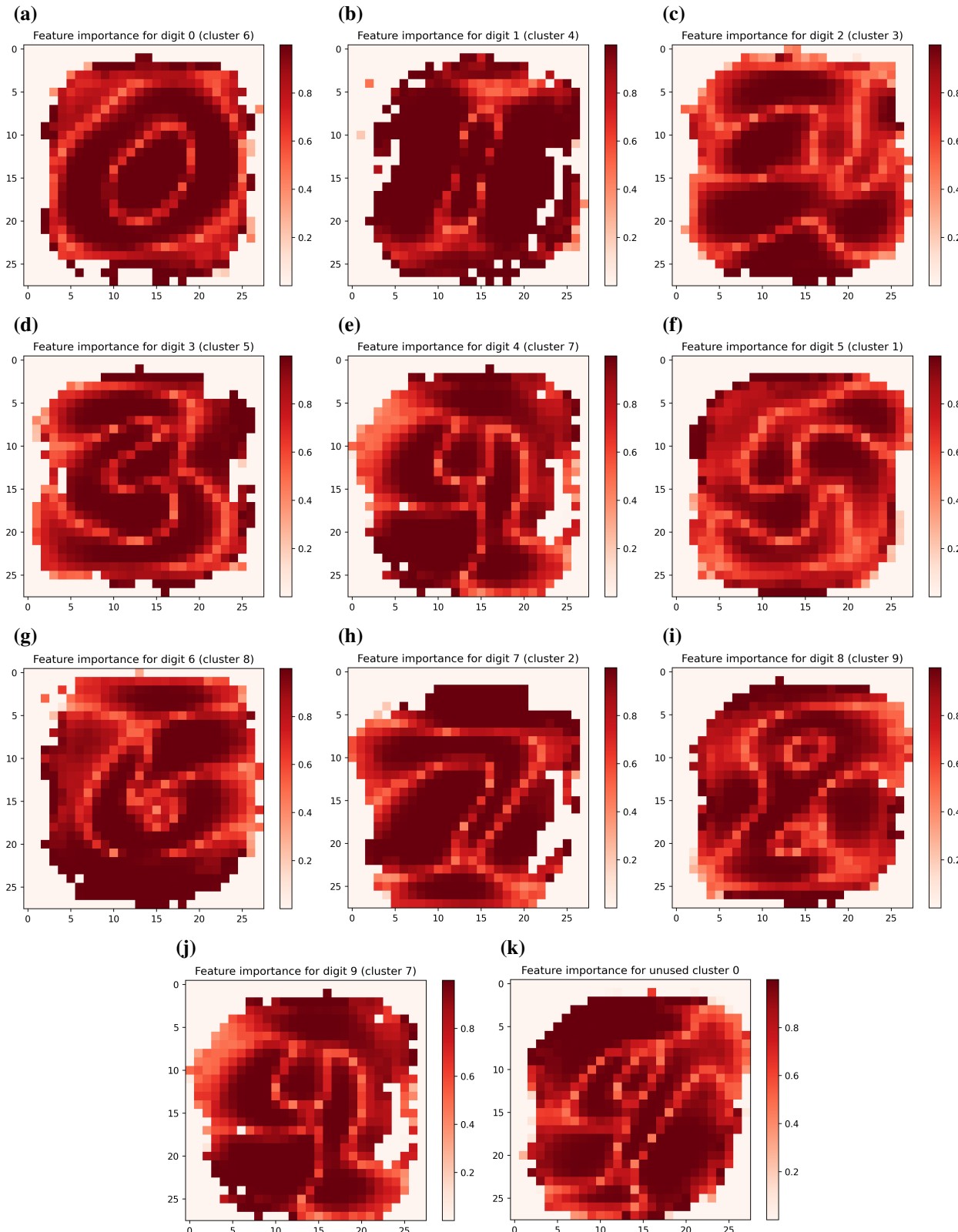

*Figure B.2.* Feature importance from the BASIL model (1+FS) trained on the MNIST 50% dataset.

*Table B.16.* NMI, Clustering Accuracy and Time for Incorrect Constraints Test of BASIL Model Aggregated over 10 Runs for DIGITS Dataset (50% Wrong Labels, Extreme Case) and the Same Test Dataset.

| | LABEL% | 20% | | | 50% | | |
|---|---|---|---|---|---|---|---|
| METRIC | MODEL | TRAIN | TEST | TIME | TRAIN | TEST | TIME |
| NMI | $(w_{nm},$FS) | $0.58\pm 0.03$ | $0.72\pm 0.02$ | $8.92\,\mathrm{m}\pm 0.37$ | $0.61\pm 0.02$ | $0.66\pm 0.05$ | $0.93\,\mathrm{h}\pm 0.05$ |
| | Gamma$(1, 2.5(10))$** | $0.60\pm 0.01$ | $\mathbf{0.75}\pm 0.02$ | $6.45\,\mathrm{m}\pm 0.42$ | $0.53\pm 0.01$ | $\mathbf{0.74}\pm 0.04$ | $0.71\,\mathrm{h}\pm 0.04$ |
| | $(w_{nm},$D+FS)* | $0.55\pm 0.03$ | $0.74\pm 0.02$ | $9.05\,\mathrm{m}\pm 0.56$ | $0.46\pm 0.04$ | $0.65\pm 0.04$ | $0.97\,\mathrm{h}\pm 0.04$ |
| | $(1,$FS) | $0.58\pm 0.02$ | $0.71\pm 0.03$ | $8.19\,\mathrm{m}\pm 0.58$ | $0.62\pm 0.01$ | $0.66\pm 0.05$ | $0.94\,\mathrm{h}\pm 0.07$ |
| | $(1,$D+FS)* | $0.58\pm 0.03$ | $0.73\pm 0.02$ | $9.19\,\mathrm{m}\pm 0.62$ | $0.59\pm 0.01$ | $0.65\pm 0.03$ | $0.95\,\mathrm{h}\pm 0.06$ |
| ACC | $(w_{nm},$FS) | $0.59\pm 0.03$ | $0.70\pm 0.03$ | $8.92\,\mathrm{m}\pm 0.37$ | $0.64\pm 0.03$ | $0.66\pm 0.06$ | $0.93\,\mathrm{h}\pm 0.05$ |
| | Gamma$(1, 2.5(10))$** | $0.64\pm 0.03$ | $\mathbf{0.74}\pm 0.05$ | $6.45\,\mathrm{m}\pm 0.42$ | $0.52\pm 0.00$ | $\mathbf{0.72}\pm 0.05$ | $0.71\,\mathrm{h}\pm 0.04$ |
| | $(w_{nm},$D+FS)* | $0.59\pm 0.02$ | $0.71\pm 0.03$ | $9.05\,\mathrm{m}\pm 0.56$ | $0.48\pm 0.03$ | $0.67\pm 0.04$ | $0.97\,\mathrm{h}\pm 0.04$ |
| | $(1,$FS) | $0.58\pm 0.02$ | $0.69\pm 0.05$ | $8.19\,\mathrm{m}\pm 0.58$ | $0.66\pm 0.03$ | $0.68\pm 0.05$ | $0.94\,\mathrm{h}\pm 0.07$ |
| | $(1,$D+FS)* | $0.59\pm 0.02$ | $0.71\pm 0.03$ | $9.19\,\mathrm{m}\pm 0.62$ | $0.63\pm 0.01$ | $0.67\pm 0.03$ | $0.95\,\mathrm{h}\pm 0.06$ |

*Table B.17.* Clustering Accuracy and Time of Models Aggregated over 10 Runs for DIGITS Dataset and the Same Test Dataset with 10% Noisy Constraints.

| | LABEL% | 20% | | | 50% | | |
|---|---|---|---|---|---|---|---|
| DATA | MODEL | TRAIN | TEST | TIME | TRAIN | TEST | TIME |
| DIGITS | MML | $0.41\pm 0.01$ | $0.49\pm 0.04$ | $45.15\,\mathrm{s}\pm 1.41$ | $0.53\pm 0.02$ | $0.59\pm 0.02$ | $3.39\,\mathrm{m}\pm 0.03$ |
| | PCK | $0.81\pm 0.02$ | $\mathbf{0.79}\pm 0.02$ | $31.96\,\mathrm{s}\pm 0.67$ | $0.86\pm 0.04$ | $0.82\pm 0.04$ | $7.99\,\mathrm{m}\pm 0.07$ |
| | MPCK | $0.69\pm 0.04$ | $0.74\pm 0.02$ | $23.52\,\mathrm{s}\pm 0.77$ | $0.71\pm 0.07$ | $0.67\pm 0.08$ | $7.23\,\mathrm{m}\pm 0.06$ |
| | DCGM | $0.49\pm 0.36$ | $0.49\pm 0.34$ | $2.24\,\mathrm{m}\pm 0.06$ | $0.45\pm 0.34$ | $0.47\pm 0.34$ | $2.30\,\mathrm{m}\pm 0.03$ |
| | BASIL | $0.81\pm 0.01$ | $\mathbf{0.79}\pm 0.03$ | $6.96\,\mathrm{m}\pm 0.28$ | $0.91\pm 0.00$ | $\mathbf{0.83}\pm 0.00$ | $0.73\,\mathrm{h}\pm 0.05$ |

*Table B.18.* Clustering Accuracy and Time of Models Aggregated over 10 Runs for DIGITS Dataset and the Same Test Dataset with 20% Noisy Constraints.

| | LABEL% | 20% | | | 50% | | |
|---|---|---|---|---|---|---|---|
| DATA | MODEL | TRAIN | TEST | TIME | TRAIN | TEST | TIME |
| DIGITS | MML | $0.40\pm 0.01$ | $0.48\pm 0.03$ | $48.34\,\mathrm{s}\pm 7.44$ | $0.48\pm 0.03$ | $0.57\pm 0.03$ | $6.19\,\mathrm{m}\pm 4.74$ |
| | PCK | $0.81\pm 0.03$ | $0.76\pm 0.01$ | $31.53\,\mathrm{s}\pm 0.20$ | $0.79\pm 0.06$ | $0.75\pm 0.05$ | $8.08\,\mathrm{m}\pm 0.03$ |
| | MPCK | $0.68\pm 0.04$ | $0.74\pm 0.02$ | $23.60\,\mathrm{s}\pm 0.05$ | $0.70\pm 0.07$ | $0.65\pm 0.06$ | $7.29\,\mathrm{m}\pm 0.05$ |
| | DCGM | $0.74\pm 0.22$ | $0.74\pm 0.22$ | $2.29\,\mathrm{m}\pm 0.05$ | $0.19\pm 0.22$ | $0.22\pm 0.22$ | $2.32\,\mathrm{m}\pm 0.03$ |
| | BASIL | $0.80\pm 0.02$ | $\mathbf{0.80}\pm 0.03$ | $6.75\,\mathrm{m}\pm 0.36$ | $0.88\pm 0.00$ | $\mathbf{0.82}\pm 0.00$ | $0.75\,\mathrm{h}\pm 0.06$ |

*Table B.19.* Clustering Accuracy and Time of Models Aggregated over 10 Runs for DIGITS Dataset and the Same Test Dataset with 30% Noisy Constraints.

| | LABEL% | 20% | | | 50% | | |
|---|---|---|---|---|---|---|---|
| DATA | MODEL | TRAIN | TEST | TIME | TRAIN | TEST | TIME |
| DIGITS | MML | $0.36\pm 0.01$ | $0.48\pm 0.03$ | $50.55\,\mathrm{s}\pm 1.65$ | $0.43\pm 0.01$ | $0.52\pm 0.04$ | $3.56\,\mathrm{m}\pm 0.06$ |
| | PCK | $0.74\pm 0.03$ | $0.77\pm 0.01$ | $30.84\,\mathrm{s}\pm 0.12$ | $0.70\pm 0.07$ | $0.69\pm 0.08$ | $7.91\,\mathrm{m}\pm 0.05$ |
| | MPCK | $0.65\pm 0.04$ | $0.75\pm 0.03$ | $22.72\,\mathrm{s}\pm 0.12$ | $0.65\pm 0.04$ | $0.62\pm 0.08$ | $7.15\,\mathrm{m}\pm 0.04$ |
| | DCGM | $0.31\pm 0.30$ | $0.33\pm 0.30$ | $2.28\,\mathrm{m}\pm 0.04$ | $0.25\pm 0.25$ | $0.29\pm 0.27$ | $2.32\,\mathrm{m}\pm 0.03$ |
| | BASIL | $0.79\pm 0.02$ | $\mathbf{0.81}\pm 0.02$ | $6.79\,\mathrm{m}\pm 0.26$ | $0.83\pm 0.00$ | $\mathbf{0.84}\pm 0.00$ | $0.72\,\mathrm{h}\pm 0.05$ |

## B.4. Additional CPRD Results

We evaluate the performance of BASIL under different parametric choices using Chebyshev similarity between training and validation cluster profiles Figure B.4, diversity of clustering, and relatedness to outcomes. For reference, we also run an unsupervised clustering variant of BASIL. Specifically, we explore:

- 10 settings of constraint weights: fixed weight values of 0.1, 1, 10, 100, and 1000; and learned weight values under

*Table B.20.* Clustering Accuracy and Time of Models Aggregated over 10 Runs for DIGITS Dataset and the Same Test Dataset with 50% Noisy Constraints.

| | LABEL% | 20% | | | 50% | | |
|---|---|---|---|---|---|---|---|
| DATA | MODEL | TRAIN | TEST | TIME | TRAIN | TEST | TIME |
| DIGITS | MML | $0.35\pm 0.02$ | $0.47\pm 0.01$ | $53.02\,\text{s}\pm 5.11$ | $0.39\pm 0.02$ | $0.49\pm 0.04$ | $4.00\,\text{m}\pm 0.61$ |
| | PCK | $0.65\pm 0.04$ | $\mathbf{0.74}\pm 0.02$ | $30.84\,\text{s}\pm 0.14$ | $0.64\pm 0.07$ | $0.70\pm 0.04$ | $7.92\,\text{m}\pm 0.07$ |
| | MPCK | $0.60\pm 0.03$ | $0.66\pm 0.03$ | $22.52\,\text{s}\pm 0.08$ | $0.60\pm 0.05$ | $0.68\pm 0.05$ | $7.17\,\text{m}\pm 0.05$ |
| | DCGM | $0.19\pm 0.15$ | $0.21\pm 0.18$ | $2.27\,\text{m}\pm 0.03$ | $0.18\pm 0.16$ | $0.22\pm 0.20$ | $2.32\,\text{m}\pm 0.04$ |
| | BASIL | $0.64\pm 0.03$ | $\mathbf{0.74}\pm 0.05$ | $6.45\,\text{m}\pm 0.42$ | $0.52\pm 0.00$ | $\mathbf{0.72}\pm 0.05$ | $0.71\,\text{h}\pm 0.04$ |

*Table B.21.* NMI, Clustering Accuracy and Time for Different BASIL Models Aggregated over 10 Runs for MNIST Dataset and the Same Test Dataset.

| | LABEL% | 20% | | | 50% | | |
|---|---|---|---|---|---|---|---|
| METRIC | MODEL | TRAIN | TEST | TIME | TRAIN | TEST | TIME |
| NMI | $(w_{nm},\text{FS})^*$ | $0.61\pm 0.01$ | $0.57\pm 0.01$ | $1.50\,\text{h}\pm 0.01$ | $0.72\pm 0.07$ | $0.62\pm 0.04$ | $1.53\,\text{h}\pm 0.02$ |
| | $(w_{nm},\text{D+FS})$ | $0.58\pm 0.03$ | $0.56\pm 0.02$ | $1.58\,\text{h}\pm 0.02$ | $0.64\pm 0.02$ | $0.59\pm 0.02$ | $1.56\,\text{h}\pm 0.01$ |
| | $(1,\text{FS})^*$ | $0.61\pm 0.01$ | $0.57\pm 0.01$ | $1.52\,\text{h}\pm 0.02$ | $0.71\pm 0.01$ | $0.62\pm 0.01$ | $1.52\,\text{h}\pm 0.02$ |
| | $(1,\text{D+FS})^{**}$ | $0.61\pm 0.01$ | $0.57\pm 0.01$ | $1.53\,\text{h}\pm 0.02$ | $0.78\pm 0.00$ | $\mathbf{0.66}\pm 0.00$ | $1.57\,\text{h}\pm 0.01$ |
| | $(1,-)$ | $0.62\pm 0.00$ | $\mathbf{0.58}\pm 0.01$ | $1.02\,\text{h}\pm 0.02$ | $0.70\pm 0.01$ | $0.61\pm 0.01$ | $1.02\,\text{h}\pm 0.01$ |
| ACC | $(w_{nm},\text{FS})^*$ | $0.73\pm 0.03$ | $0.68\pm 0.02$ | $1.50\,\text{h}\pm 0.01$ | $0.81\pm 0.09$ | $0.73\pm 0.07$ | $1.53\,\text{h}\pm 0.02$ |
| | $(w_{nm},\text{D+FS})$ | $0.68\pm 0.04$ | $0.64\pm 0.04$ | $1.58\,\text{h}\pm 0.02$ | $0.73\pm 0.04$ | $0.68\pm 0.03$ | $1.56\,\text{h}\pm 0.01$ |
| | $(1,\text{FS})^*$ | $0.73\pm 0.03$ | $0.68\pm 0.02$ | $1.52\,\text{h}\pm 0.02$ | $0.79\pm 0.03$ | $0.71\pm 0.02$ | $1.52\,\text{h}\pm 0.02$ |
| | $(1,\text{D+FS})^{**}$ | $0.73\pm 0.01$ | $0.67\pm 0.01$ | $1.53\,\text{h}\pm 0.02$ | $0.89\pm 0.00$ | $\mathbf{0.79}\pm 0.00$ | $1.57\,\text{h}\pm 0.01$ |
| | $(1,-)$ | $0.74\pm 0.01$ | $\mathbf{0.69}\pm 0.01$ | $1.02\,\text{h}\pm 0.02$ | $0.77\pm 0.03$ | $0.70\pm 0.03$ | $1.02\,\text{h}\pm 0.01$ |

*Table B.22.* NMI, Clustering Accuracy and Time for Incorrect Constraints Test of BASIL Model Aggregated over 10 Runs for MNIST Dataset (20% Wrong Labels) and the Same Test Dataset.

| | LABEL% | 20% | | | 50% | | |
|---|---|---|---|---|---|---|---|
| METRIC | MODEL | TRAIN | TEST | TIME | TRAIN | TEST | TIME |
| NMI | $(w_{nm},\text{FS})$ | $0.58\pm 0.01$ | $\mathbf{0.56}\pm 0.01$ | $1.50\,\text{h}\pm 0.02$ | $0.46\pm 0.01$ | $0.56\pm 0.01$ | $1.53\,\text{h}\pm 0.02$ |
| | $\text{Gamma}(1, 1.2)^*$ | $0.57\pm 0.02$ | $\mathbf{0.56}\pm 0.02$ | $1.54\,\text{h}\pm 0.01$ | $0.50\pm 0.02$ | $0.58\pm 0.02$ | $1.52\,\text{h}\pm 0.02$ |
| | $(w_{nm},\text{D+FS})$ | $0.55\pm 0.01$ | $0.54\pm 0.01$ | $1.58\,\text{h}\pm 0.02$ | $0.50\pm 0.02$ | $0.54\pm 0.02$ | $1.56\,\text{h}\pm 0.02$ |
| | $(1,\text{FS})^*$ | $0.59\pm 0.01$ | $\mathbf{0.56}\pm 0.01$ | $1.52\,\text{h}\pm 0.02$ | $0.50\pm 0.02$ | $0.58\pm 0.02$ | $1.52\,\text{h}\pm 0.02$ |
| | $(1,\text{D+FS})^*$ | $0.57\pm 0.03$ | $0.53\pm 0.02$ | $1.53\,\text{h}\pm 0.02$ | $0.73\pm 0.00$ | $0.62\pm 0.00$ | $1.57\,\text{h}\pm 0.01$ |
| | $(w_{nm}^*,\text{D}^*\text{+FS})^{**}$ | $0.60\pm 0.02$ | $\mathbf{0.56}\pm 0.02$ | $1.54\,\text{h}\pm 0.01$ | $0.70\pm 0.00$ | $\mathbf{0.63}\pm 0.00$ | $1.27\,\text{h}\pm 0.03$ |
| ACC | $(w_{nm},\text{FS})$ | $0.68\pm 0.03$ | $0.65\pm 0.02$ | $1.50\,\text{h}\pm 0.02$ | $0.58\pm 0.01$ | $0.64\pm 0.02$ | $1.53\,\text{h}\pm 0.02$ |
| | $\text{Gamma}(1, 1.2)^*$ | $0.71\pm 0.03$ | $\mathbf{0.66}\pm 0.03$ | $1.54\,\text{h}\pm 0.01$ | $0.62\pm 0.02$ | $0.66\pm 0.02$ | $1.52\,\text{h}\pm 0.02$ |
| | $(w_{nm},\text{D+FS})$ | $0.65\pm 0.02$ | $0.62\pm 0.02$ | $1.58\,\text{h}\pm 0.02$ | $0.61\pm 0.02$ | $0.62\pm 0.02$ | $1.56\,\text{h}\pm 0.02$ |
| | $(1,\text{FS})^*$ | $0.71\pm 0.02$ | $\mathbf{0.66}\pm 0.02$ | $1.52\,\text{h}\pm 0.02$ | $0.62\pm 0.03$ | $0.66\pm 0.03$ | $1.52\,\text{h}\pm 0.02$ |
| | $(1,\text{D+FS})^*$ | $0.66\pm 0.04$ | $0.61\pm 0.03$ | $1.53\,\text{h}\pm 0.02$ | $0.86\pm 0.00$ | $\mathbf{0.76}\pm 0.00$ | $1.57\,\text{h}\pm 0.01$ |
| | $(w_{nm}^*,\text{D}^*\text{+FS})^{**}$ | $0.71\pm 0.03$ | $\mathbf{0.66}\pm 0.03$ | $1.54\,\text{h}\pm 0.01$ | $0.84\pm 0.00$ | $\mathbf{0.76}\pm 0.00$ | $1.27\,\text{h}\pm 0.03$ |

*Table B.23.* Component synergy on DIGITS at 20% supervision. Adaptive constraint weighting ($w_{nm}$) yields gains in both clustering accuracy and constraint-reliability AUROC only when paired with feature selection (FS). Configurations with fixed $w_{nm}=1$ have AUROC $= 0.500$ by design (no per-constraint weight to threshold). Aggregated over 10 random seeds.

| | TEST ACC | | | AUROC | |
|---|---|---|---|---|---|
| CONFIGURATION | CLEAN | 20% NOISE | 50% NOISE | 20% NOISE | 50% NOISE |
| $(1,-)$ | $0.80\pm 0.00$ | $0.76\pm 0.02$ | $0.68\pm 0.01$ | $0.500$ | $0.500$ |
| $(1, \text{FS})$ | $0.80\pm 0.02$ | $0.76\pm 0.04$ | $0.69\pm 0.00$ | $0.500$ | $0.500$ |
| $(w_{nm},-)$ | $0.76\pm 0.02$ | $0.75\pm 0.02$ | $0.68\pm 0.01$ | $0.630$ | $0.802$ |
| $(w_{nm}, \text{FS})$ | $0.78\pm 0.04$ | $\mathbf{0.80}\pm 0.03$ | $\mathbf{0.74}\pm 0.05$ | $\mathbf{0.953}$ | $\mathbf{0.914}$ |

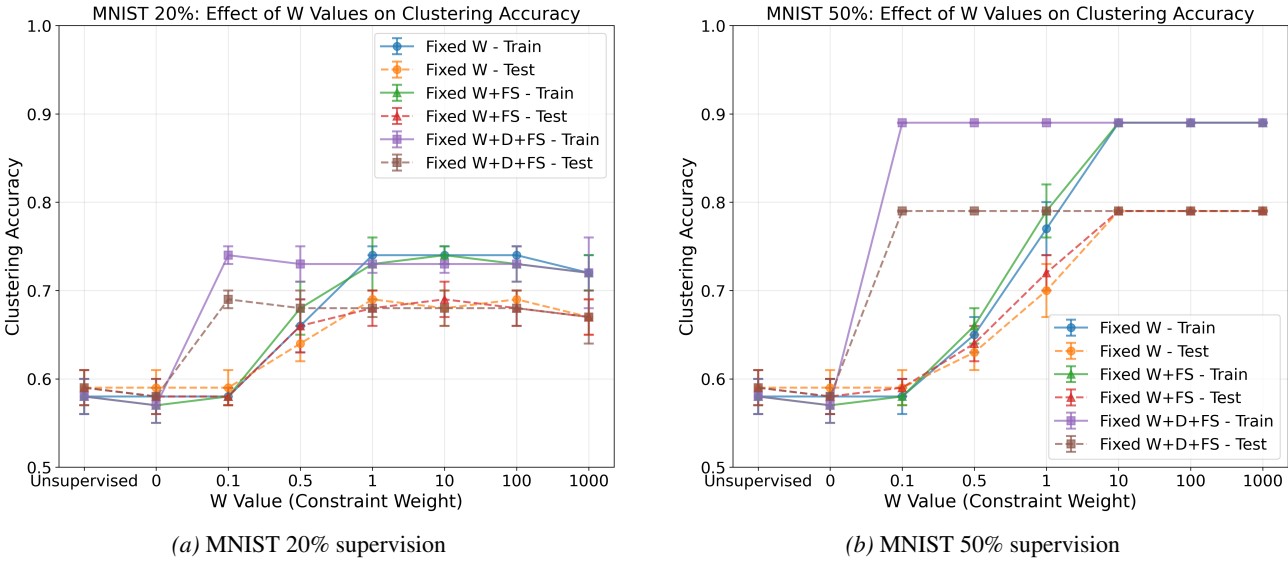

*(a)* MNIST 20% supervision    *(b)* MNIST 50% supervision

*Figure B.3.* Comparison of test clustering accuracy across different BASIL model configurations with varying fixed $w$ values (SE computed from 10 runs).

prior beta parameter of 0.1, 0.5, 1, 3, 5;

- 3 different constraint sampling schemes: fully random; balanced, such that equal number of positive and negative constraints are sampled from each outcome label or pair of outcome labels; doubly balanced, which in addition to balancing among possible labels / pairs, balances the number of positive and negative constraints;

- 6 set values of number of clusters: 8, 12, 16, 20, 24, 28.

We evaluate diversity as the average Chebyshev distance between all non-repeating pairs of clusters. We quantify the relatedness to outcomes as the proportion of clusters with standardised outcome ratio lower than 0.5, or higher than 2. Standardised outcome ratio is the ratio of prevalence of the outcome in the cluster vs. the prevalence in the respective split.

In terms of stability, we observe that weights fixed to a value of 1 or higher lead to poor alignment between training and validation, as well as a degradation of alignment with growing number of clusters (Figure B.4). In contrast, when weights are learned, the alignment remains stable, robust to sampling choices, and does not drop as the number of clusters is increased. In all cases, unsupervised clustering remains the most stable and well aligned.

When taking into account how diverse the clusters are in terms of structure, and how related to the outcome, the supervision leads to better diversity of clusters (Figure B.5). Further, for the mortality outcome the supervision of the constraints also improves the outcome-relatedness of clusters, particularly noticeable for the 28 clusters solution.

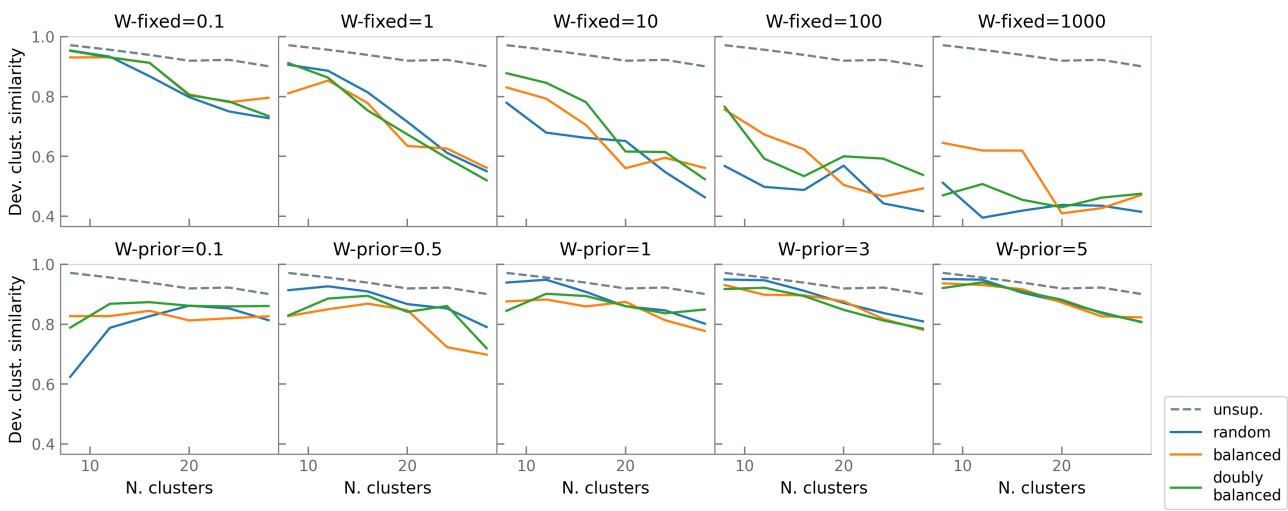

*Figure B.4.* Relationship between the number of clusters and stability of clustering, at different parametric choices.

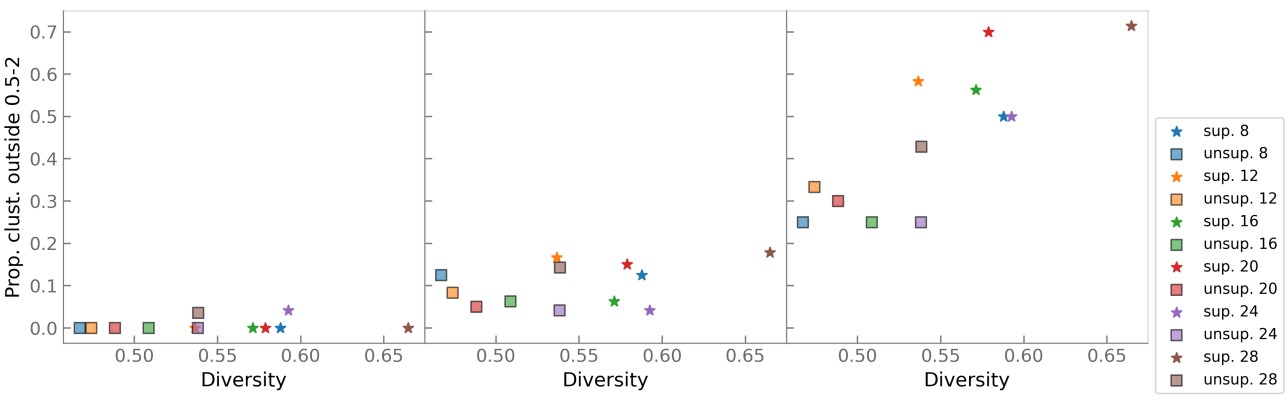

*Figure B.5.* Comparison of unsupervised and supervised clustering in terms of the diversity of a clustering solution, and relatedness to outcomes, for different number of clusters and selected parametric setting of the supervised method (learned weights, beta prior 1, balanced sampling); validation split.

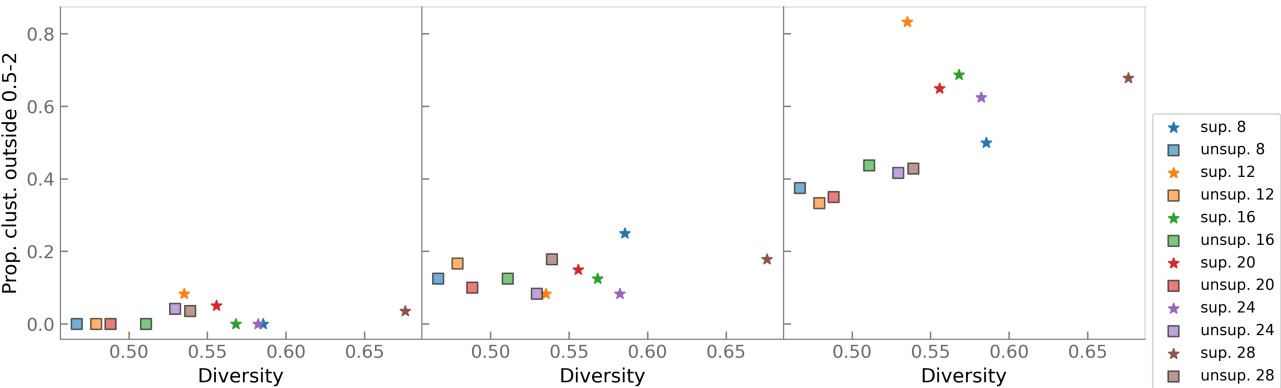

*Figure B.6.* Comparison of unsupervised and supervised clustering in terms of the diversity of a clustering solution, and relatedness to outcomes, for different number of clusters and selected parametric setting of the supervised method (learned weights, beta prior 1, balanced sampling); test split.

## C. Additional Information on Stochastic Variational Inference

### C.1. Joint Probability Factorization

The joint probability of $X$, $\mathbf{z}$ and $\boldsymbol{\Theta}$ in the HMRF framework can be factorized as follows,

$$
\begin{aligned}
p(X, \mathbf{z}, \boldsymbol{\Theta} \mid Y) &= p(X \mid \mathbf{z}, \boldsymbol{\Theta}) p(\mathbf{z} \mid \boldsymbol{\Theta}, Y) p(\boldsymbol{\Theta}) \\
&= \prod_{n=1}^{N} p(\mathbf{x}_n \mid z_n, \boldsymbol{\theta}_k) p(\mathbf{z} \mid \boldsymbol{\pi}, W, \bar{W}, Y) p(W) p(\bar{W}) \\
&\quad \prod_{k=1}^{K} p(\boldsymbol{\theta}_k) p(\boldsymbol{\gamma}_k) p(\boldsymbol{\pi}).
\end{aligned}
\tag{C.15}
$$

### C.2. General SVI Update Rule

Assuming that the complete conditional distributions are in the exponential family and that the global parameters follow conjugate priors, the general update rule for a global variational parameter $\boldsymbol{\nu}$ at iteration $t$ is given by:

$$
\boldsymbol{\nu}^{(t)} = (1 - \rho_t)\boldsymbol{\nu}^{(t-1)} + \rho_t \left( \boldsymbol{\nu}_0 + \frac{N}{M} \sum_{n \in \mathcal{B}_t} \hat{s}_n \right),
\tag{C.16}
$$

where $\rho_t$ is the learning rate, $\mathcal{B}_t$ is a mini-batch of size $M$, $\hat{s}_n$ denotes the sufficient statistics contributed by data point $n$, and $\boldsymbol{\nu}_0$ corresponds to the natural parameters of the conjugate prior.

### C.3. Variational Inference Background

In Variational Inference (VI), we want to approximate the posterior $p(\mathbf{z}, \boldsymbol{\Theta})$ by the variational distribution $q(\mathbf{z}, \boldsymbol{\Theta} \mid \Phi, \boldsymbol{\nu}) = q(\boldsymbol{\Theta}) \prod_{n=1}^{N} q(z_n)$ using mean-field factorization where $\Phi$ is the variational parameters for $\mathbf{z}$ representing the probability matrix and $\boldsymbol{\nu}$ is the variational parameters for $\boldsymbol{\Theta}$. Here $\mathbf{z} = \{z_n\}_{n=1}^{N}$ are the local latent variables and $\boldsymbol{\Theta}$ are global latent variables containing all parameters related to latent clusters.

The optimization objective is the Evidence Lower Bound (ELBO):

$$
\mathcal{L}(q) = \mathbb{E}_q[\log p(X, \mathbf{z}, \boldsymbol{\Theta})] - \mathbb{E}_q[\log q(\mathbf{z}, \boldsymbol{\Theta})].
$$

Maximizing $\mathcal{L}(q)$ minimizes the KL-divergence $\mathrm{KL}(q \,\|\, p)$.

In the Coordinate Ascent algorithm in standard VI, we maximize $\mathcal{L}(q)$ w.r.t. each factor while keeping others fixed. Taking the derivative and setting it to zero yields: $\log q^*(z_n) = \mathbb{E}_{-z_n}[\log p(X, \mathbf{z}, \boldsymbol{\Theta})] + \mathrm{const}$, and similarly for $q(\boldsymbol{\Theta})$. The natural gradient of conjugate exponential families $\tilde{\nabla}_{\boldsymbol{\nu}} \mathcal{L} = 0$ gives the coordinate ascent update for global parameters:

$$
\boldsymbol{\nu} = \boldsymbol{\nu}_0 + s = \boldsymbol{\nu}_0 + \sum_{n=1}^{N} \mathbb{E}_q[t(\mathbf{x}_n, z_n)]
\tag{C.17}
$$

where $\boldsymbol{\nu}_0$ is the natural parameter of the prior, and $t(\mathbf{x}_n, z_n)$ are sufficient statistics from the local latent variables. In addition, we update the local latent variable:

$$
\Phi_{nk} \propto \exp \left\{ \mathbb{E}_q[\log p(\mathbf{x}_n \mid z_n = k, \boldsymbol{\theta}_k)]) + \mathbb{I}(G_n = \emptyset)\mathbb{E}_q[\log \pi_k] - \sum_{m \in G_n} \sum_{j=1}^{K} \Phi_{mj} \mathcal{V}(k, j) \right\}.
\tag{C.18}
$$

where $\mathcal{V}(k, j) = \mathcal{V}_m(k, j)$ when $y_{nm} = 1$ and $\mathcal{V}(k, j) = \mathcal{V}_c(k, j)$ when $y_{nm} = -1$.

The optimal global variational parameter in standard VI equals the prior plus the expected sufficient statistics from all data. In SVI, it replaces this with a stochastic estimate based on a mini-batch $\mathcal{B}_t$ of size $M$: $\hat{s}_t = \frac{N}{M} \sum_{n \in \mathcal{B}_t} \hat{s}_n = \frac{N}{M} \sum_{n \in \mathcal{B}_t} \mathbb{E}_q[t(\mathbf{x}_n, z_n)]$. Then the Coordinate Ascent VI (CAVI) update is relaxed into a stochastic natural gradient update at each iteration $t$:

$$
\boldsymbol{\nu}^{(t)} = (1 - \rho_t)\boldsymbol{\nu}^{(t-1)} + \rho_t(\boldsymbol{\nu}_0 + \hat{s}_t)
\tag{C.19}
$$

where $\rho_t$ is the learning rate with decay $\rho_t = (\tau_0 + t)^{-\kappa}$. For the local latent variable $\Phi$, the update formula remains the same; however, we compute local updates only for these $M$ samples at each iteration. The total computation complexity is then $O(MKD)$ instead of $O(NKD)$ at each iteration.

### C.4. Global Parameters Independent of Likelihood

The CAVI update for variational parameters of $\boldsymbol{\pi}$ is independent of likelihood and feature selection, so it is always unchanged. If $q(\boldsymbol{\pi}) = \text{Dir}(\boldsymbol{\pi} \mid \boldsymbol{\alpha})$ and we have

$$\alpha_k = \alpha_0 + \sum_{n=1}^{N} \Phi_{nk} \tag{C.20}$$

### C.5. Local Parameters Independent of Likelihood

In addition, the variational parameters of constraint violation cost $w_{nm}$ and $\bar{w}_{nm}$ are also independent of likelihood and feature selection. If $q(w_{nm}) = \text{Gamma}(w_{nm} \mid \beta_{nm,1}, \beta_{nm,2})$ and $q(\bar{w}_{nm}) = \text{Gamma}(\bar{w}_{nm} \mid \bar{\beta}_{nm,1}, \bar{\beta}_{nm,2})$, the related terms in the ELBO is

$$\log q(w_{nm}) \propto \log p(w_{nm}) - \sum_{z_m \in G_n} \mathbb{E}_q[\mathcal{V}(z_n, z_m \mid y_{nm} = 1)]$$

$$\propto (\beta_{0,1} - 1) \log w_{nm} - \left\{ \beta_{0,2} + \mathbb{I}(m \in G_n, y_{nm} = 1) \sum_{j=1}^{K} \sum_{k=1}^{K} \Phi_{nj} \Phi_{mk} \mathbb{D}_{\text{KL}}^{\text{sym}} \right\} w_{nm} \tag{C.21}$$

so, $\beta_{nm,1} = \beta_{0,1}$ and $\beta_{nm,2} = \beta_{0,2} + \mathbb{I}(m \in G_n, y_{nm} = 1) \sum_{j=1}^{K} \sum_{k=1}^{K} \Phi_{nj} \Phi_{mk} \mathbb{D}_{\text{KL}}^{\text{sym}}$. Similarly,

$$\log q(\bar{w}_{nm}) \propto \log p(\bar{w}_{nm}) - \sum_{z_m \in G_n} \mathbb{E}_q[\mathcal{V}(z_n, z_m \mid y_{nm} = -1)]$$

$$\propto (\bar{\beta}_{0,1} - 1) \log \bar{w}_{nm} - \left\{ \bar{\beta}_{0,2} + \mathbb{I}(m \in G_n, y_{nm} = -1) \sum_{k=1}^{K} \Phi_{nk} \Phi_{mk} \mathcal{V}^{\text{max}} \right\} \bar{w}_{nm} \tag{C.22}$$

so, $\bar{\beta}_{nm,1} = \bar{\beta}_{0,1}$ and $\bar{\beta}_{nm,2} = \bar{\beta}_{0,2} + \mathbb{I}(m \in G_n, y_{nm} = -1) \sum_{k=1}^{K} \Phi_{nk} \Phi_{mk} \mathcal{V}^{\text{max}}$. We have $\mathbb{E}[w_{nm}] = \frac{\beta_{nm,1}}{\beta_{nm,2}}$ and $\mathbb{E}[\bar{w}_{nm}] = \frac{\bar{\beta}_{nm,1}}{\bar{\beta}_{nm,2}}$.

### C.6. SVI Implementation for Multivariate Bernoulli Likelihood

We adapt the method in Echraibi et al. (2019) to the multivariate Bernoulli distribution of samples. The only change is the likelihood distribution and corresponding parameters,

$$\theta_{kd} \sim \text{Beta}(a_0, b_0),$$

$$\mathbf{x}_n \mid z_n = k, \boldsymbol{\theta}_k \sim \prod_{d=1}^{D} \text{Bern}(x_{nd} \mid \theta_{kd}), \tag{C.23}$$

with mean-field VI approximation of $\boldsymbol{\theta}_k$ being $q(\theta_{kd}) = \text{Beta}(\theta_{kd} \mid a_{kd}, b_{kd})$ and $q(\boldsymbol{\theta}_k) = \prod_{d=1}^{D} \text{Beta}(\theta_{kd} \mid a_{kd}, b_{kd})$. Thus,

$$\mathbb{E}_q[\log p(\mathbf{x}_n \mid z_n = k, \boldsymbol{\theta}_k)] = \sum_{d=1}^{D} \left\{ x_{nd} \mathbb{E}_q[\log \theta_{kd}] + (1 - x_{nd}) \mathbb{E}_q[\log(1 - \theta_{kd})] \right\}$$

$$= \sum_{d=1}^{D} \left\{ x_{nd}(\psi(a_{kd}) - \psi(a_{kd} + b_{kd})) + (1 - x_{nd})(\psi(b_{kd}) - \psi(a_{kd} + b_{kd})) \right\}. \tag{C.24}$$

where $\psi$ is the digamma function. The CAVI update formula for variational parameters of likelihood parameters $(a_{kd}, b_{kd})$ is straightforward by applying conjugate priors by Equation (C.17), that is, $a_{kd} = a_0 + \sum_{n=1}^{N} \Phi_{nk} x_{nd}$ and $b_{kd} = b_0 + \sum_{n=1}^{N} \Phi_{nk}(1 - x_{nd})$. This can be easily extended to the stochastic natural gradient update in SVI using Equation (C.19).

The potential matrix by Echraibi et al. (2019) is $\mathbb{D}_{\mathrm{KL}}^{\mathrm{sym}}[q(\boldsymbol{\theta}_k) \,\|\, q(\boldsymbol{\theta}_l)] = \frac{1}{2}\{\sum_{d=1}^{D}\{\log \frac{B(a_{ld},b_{ld})}{B(a_{kd},b_{kd})} + (a_{kd} - a_{ld})[\psi(a_{kd}) - \psi(a_{kd}+b_{kd})] + (b_{kd}-b_{ld})[\psi(b_{kd})-\psi(a_{kd}+b_{kd})]\} + \sum_{d=1}^{D}\{\log \frac{B(a_{kd},b_{kd})}{B(a_{ld},b_{ld})} + (a_{ld}-a_{kd})[\psi(a_{ld})-\psi(a_{ld}+b_{ld})] + (b_{ld}-b_{kd})[\psi(b_{ld})-\psi(a_{ld}+b_{ld})]\}\}$, where $B(\cdot,\cdot)$ is the Beta function.

Our proposed potential matrix is

$$\mathbb{D}_{\mathrm{KL}}^{\mathrm{sym}}[p(\mathbf{x}\,|\,\boldsymbol{\theta}_k)\,\|\,p(\mathbf{x}\,|\,\boldsymbol{\theta}_l)] = \frac{1}{2}\sum_{d=1}^{D}(\mathbb{E}_q[\theta_{kd}] - \mathbb{E}_q[\theta_{ld}])\{(\mathbb{E}_q[\log\theta_{kd}] - \mathbb{E}_q[\log\theta_{ld}]) - (\mathbb{E}_q[\log(1-\theta_{kd})] - \mathbb{E}_q[\log(1-\theta_{ld})])\}$$

which is for posterior data distribution. This will change to new component parameters by feature selection described below. Then the probability matrix $\Phi$ is updated using Equation (C.18) by plugging in Equation (C.24) and these potentials.

**Feature Selection Adaptation**    When incorporating the feature selection that introduces feature importance weights $\gamma_{kd}$, the corresponding variational distribution are $q(\gamma_{kd}) = \mathrm{Beta}(\gamma_{kd}\,|\,c_{kd}, d_{kd})$. We need to change to the new component parameters after weighting for all computing. Then for the Bernoulli likelihood,

$$\mathbb{E}_q[\log p(\mathbf{x}_n\,|\,z_n = k, \tilde{\boldsymbol{\theta}}_k)] = \sum_{d=1}^{D}\left\{x_{nd}\mathbb{E}_q[\log\tilde{\theta}_{kd}] + (1-x_{nd})\mathbb{E}_q[\log(1-\tilde{\theta}_{kd})]\right\}$$

where $\mathbb{E}_q[\log\tilde{\theta}_{kd}] = \mathbb{E}_q[\gamma_{kd}]\mathbb{E}_q[\log\theta_{kd}] + (1-\mathbb{E}_q[\gamma_{kd}])\log\theta_{0d}$ and $\mathbb{E}_q[\log(1-\tilde{\theta}_{kd})] = \mathbb{E}_q[\gamma_{kd}]\mathbb{E}_q[\log(1-\theta_{kd})] + (1-\mathbb{E}_q[\gamma_{kd}])\log(1-\theta_{0d})$ by the first-order expectation approximation. For the likelihood parameters $\theta_{kd}$, the update is still conjugate, but with a feature importance weighting, that is

$$a_{kd} = a_0 + \mathbb{E}_q[\gamma_{kd}]\sum_{n=1}^{N}\Phi_{nk}x_{nd}$$

$$b_{kd} = b_0 + \mathbb{E}_q[\gamma_{kd}]\sum_{n=1}^{N}\Phi_{nk}(1-x_{nd})$$

where $\mathbb{E}_q[\gamma_{kd}] = \frac{c_{kd}}{c_{kd}+d_{kd}}$.

*Proof.* The terms in the ELBO that depend on $\theta_{kd}$ are

$$\mathcal{L}(a_{kd}, b_{kd}) = \mathbb{E}_q[\log p(\theta_{kd})] + \sum_n \mathbb{E}_q[\Phi_{nk}\log p(x_{nd}\,|\,z_n = k, \tilde{\theta}_{kd})] - \mathbb{E}_q[\log q(\theta_{kd})]$$

$$= \sum_n \Phi_{nk}\left(x_{nd}\mathbb{E}_q[\log\tilde{\theta}_{kd}] + (1-x_{nd})\mathbb{E}_q[\log(1-\tilde{\theta}_{kd})]\right)$$

$$+ (a_0 - a_{kd})\mathbb{E}_q[\log\theta_{kd}] + (b_0 - b_{kd})\mathbb{E}_q[\log(1-\theta_{kd})] + \log B(a_{kd}, b_{kd}) - \log B(a_0, b_0)$$

$$= \sum_n \Phi_{nk}\{x_{nd}\mathbb{E}_q[\gamma_{kd}](\mathbb{E}_q[\log\theta_{kd}] - \log\theta_{0d}) + (1-x_{nd})\mathbb{E}_q[\gamma_{kd}](\mathbb{E}_q[\log(1-\theta_{kd})] - \log(1-\theta_{0d}))\}$$

$$+ (a_0 - a_{kd})\mathbb{E}_q[\log\theta_{kd}] + (b_0 - b_{kd})\mathbb{E}_q[\log(1-\theta_{kd})] + \log B(a_{kd}, b_{kd}) + \mathrm{const}$$

$$= \mathbb{E}_q[\gamma_{kd}]l_{kd} + +(a_0 - a_{kd})\mathbb{E}_q[\log\theta_{kd}] + (b_0 - b_{kd})\mathbb{E}_q[\log(1-\theta_{kd})] + \log B(a_{kd}, b_{kd}) + \mathrm{const}$$

where $l_{kd} = \sum_n \Phi_{nk}\{x_{nd}(\mathbb{E}_q[\log\theta_{kd}] - \log\theta_{0d}) + (1-x_{nd})(\mathbb{E}_q[\log(1-\theta_{kd})] - \log(1-\theta_{0d}))\}$. When taking the derivative w.r.t $a_{kd}$ and $b_{kd}$, the only change compared to the original derivation is a weighting $\mathbb{E}_q[\gamma_{kd}]$ around sufficient statistics. □

For the update rules of variational parameters of $\gamma_{kd}$, we have similar terms in ELBO by changing the prior and variational distribution related to $\gamma_{kd}$. Continuing on the above derivation, we take derivative w.r.t $c_{kd}$ and $d_{kd}$ respectively and yield:

$$\frac{\partial\mathcal{L}}{\partial c_{kd}} = (\boldsymbol{\lambda}_{\gamma 1} - c_{kd})\psi_1(c_{kd}) + (c_{kd} + d_{kd} - (\boldsymbol{\lambda}_{\gamma 1} + \boldsymbol{\lambda}_{\gamma 2}))\psi_1(c_{kd} + d_{kd}) + l_{kd}\frac{d_{kd}}{(c_{kd} + d_{kd})^2},$$

$$\frac{\partial\mathcal{L}}{\partial d_{kd}} = (\boldsymbol{\lambda}_{\gamma 2} - d_{kd})\psi_1(d_{kd}) + (c_{kd} + d_{kd} - (\boldsymbol{\lambda}_{\gamma 1} + \boldsymbol{\lambda}_{\gamma 2}))\psi_1(c_{kd} + d_{kd}) - l_{kd}\frac{c_{kd}}{(c_{kd} + d_{kd})^2}, \quad \text{(C.25)}$$

where $\psi_1$ is the trigamma function. Near a stationary point, the trigamma terms balance and the $l_{kd}$-bearing terms drive the residual gradient direction. When $l_{kd} > 0$, $\mathbb{E}_q[\gamma_{kd}]$ increases and when $l_{kd} < 0$, $\mathbb{E}_q[\gamma_{kd}]$ decreases. We therefore define the approximated update rule as:

$$c_{kd} = \boldsymbol{\lambda}_{\gamma 1} + \max(0, l_{kd})$$
$$d_{kd} = \boldsymbol{\lambda}_{\gamma 2} + \max(0, -l_{kd}). \tag{C.26}$$

### C.7. Predictive Posterior Probability

Given a new data $X^{\text{new}}$, it is of interest to know its posterior probability of cluster memberships from the trained model i.e. $\Phi_{\text{new},k} = p(z_{\text{new}} \mid \mathbf{x}_{\text{new}}, \mathcal{M})$. Given the variational approximation, we marginalize over the parameter posteriors $q(\boldsymbol{\pi}, \{\boldsymbol{\theta}_k, \boldsymbol{\gamma}_k\}_{k=1}^K)$, that is

$$\Phi_{\text{new},k} \propto \mathbb{E}_{q(\pi_k)q(\boldsymbol{\theta}_k)q(\boldsymbol{\gamma}_k)}\left[p(\mathbf{x}_{\text{new}} \mid z_{\text{new}} = k, \tilde{\boldsymbol{\theta}}_k)p(z_{\text{new}} = k)\right]. \tag{C.27}$$

We approximate it by replacing parameters with their posterior expectations.

**For Bernoulli Distribution**  Equation (C.27) becomes:

$$\log \Phi_{\text{new},k} \propto \mathbb{E}_q\left[\log p(\mathbf{x}_{\text{new}} \mid z_{\text{new}} = k, \tilde{\boldsymbol{\theta}}_k)\right] + \mathbb{E}_q[\log \pi_k]$$

$$\propto \mathbb{E}_q[\log \pi_k] + \sum_{d=1}^D \mathbb{E}_q[\gamma_{kd}]\left\{x_{\text{new},d}(\mathbb{E}_q[\log \theta_{kd}] - \log \theta_{0d}) + (1 - x_{\text{new},d})(\mathbb{E}_q[\log(1 - \theta_{kd})] - \log(1 - \theta_{0d}))\right\}$$

## D. Convergence Analysis of SVI for BASIL

We provide a theoretical analysis of the convergence properties of the SVI algorithm for the full BASIL model of Section 4, including the multivariate Bernoulli likelihood with Beta priors, adaptive constraint weights $w_{nm}, \bar{w}_{nm}$ with Gamma priors, feature-relevance variables $\gamma_{kd}$ with Beta priors, and the cluster-divergence KL potential (Equation (7)). The mean-field variational family is

$$q(\mathbf{z}, \boldsymbol{\Theta}) = q(\boldsymbol{\pi}) \prod_{k=1}^K q(\boldsymbol{\theta}_k)\, q(\boldsymbol{\gamma}_k) \prod_{(n,m)} q(w_{nm})\, q(\bar{w}_{nm}) \prod_{n=1}^N q(z_n).$$

Following the categorisation of update rules in Section 5, the local variables $w_{nm}, \bar{w}_{nm}$ follow exact conjugate coordinate-ascent steps given the current assignments $\Phi$ (Gamma posteriors, Equations (C.21) and (C.22)), while the global variables $\boldsymbol{\nu} = \{\boldsymbol{\epsilon}, \{a_k, b_k\}, \{c_k, d_k\}\}$ are updated by SVI. We analyse the convergence of the SVI updates over $\boldsymbol{\nu}$ on the approximate ELBO $\tilde{\mathcal{L}}$ that uses the first-order expectation approximation for $\mathbb{E}_q[\log \tilde{\theta}_{kd}]$ in the feature-relevance update.

### D.1. Assumptions

We require the following regularity conditions:

**Assumption D.1** (Step Size Schedule)**.**  The learning rate sequence $\{\rho_t\}_{t \geq 1}$ satisfies the Robbins-Monro conditions:

$$\sum_{t=1}^\infty \rho_t = \infty \quad \text{and} \quad \sum_{t=1}^\infty \rho_t^2 < \infty.$$

A standard choice is $\rho_t = (\tau + t)^{-\kappa}$ with $\kappa \in (0.5, 1]$ and delay $\tau > 0$.

**Assumption D.2** (Bounded Constraint Graph)**.**  The constraint neighborhood is uniformly bounded: $|G_n| \leq g_{\max}$ for all $n \in \{1, \ldots, N\}$, where $g_{\max}$ is a fixed constant independent of $N$.

**Assumption D.3** (Bounded Potentials)**.**  The KL-divergence potential is bounded: there exists $V_{\max} < \infty$ such that

$$\mathbb{D}_{\text{KL}}^{\text{sym}}[p(\mathbf{x} \mid \boldsymbol{\theta}_k) \,\|\, p(\mathbf{x} \mid \boldsymbol{\theta}_j)] \leq V_{\max}$$

for all $k, j \in \{1, \ldots, K\}$ and all valid parameter configurations.

## D.2. Main Results

**Proposition D.4** (Gradient Estimator Properties). *Partition the global parameter vector as $\boldsymbol{\nu} = (\boldsymbol{\nu}_c, \boldsymbol{\nu}_h)$, where $\boldsymbol{\nu}_c = \{\boldsymbol{\epsilon}, (a_{kd}, b_{kd})\}$ is the exact-conjugate block and $\boldsymbol{\nu}_h = \{(c_{kd}, d_{kd})\}$ is the approximate-CVI block. For a uniformly sampled mini-batch $\mathcal{B}_t$ of size $M$:*

*(a) The conjugate-block stochastic natural gradient is unbiased,*

$$\mathbb{E}_{\mathcal{B}_t}\left[\tilde{\nabla}_{\boldsymbol{\nu}_c}\hat{\tilde{\mathcal{L}}}_t\right] = \tilde{\nabla}_{\boldsymbol{\nu}_c}\tilde{\mathcal{L}}.$$

*(b) The CVI-block estimator is sign-aligned with the exact CVI natural gradient (under Theorem D.3) but biased by the trigamma factor $d_{kd}/(c_{kd} + d_{kd})^2$ that the non-negative truncation in Equation (C.26) omits. Theorem D.6(b) quantifies the resulting stationary-point displacement.*

*Here $\tilde{\mathcal{L}}$ differs from the exact ELBO only through the first-order approximation $\mathbb{E}_q[\log \tilde{\theta}_{kd}] \approx \mathbb{E}_q[\gamma_{kd}]\mathbb{E}_q[\log \theta_{kd}] + (1 - \mathbb{E}_q[\gamma_{kd}]) \log \theta_{0d}$ used in the feature-relevance update.*

*Proof.* **(a)** The mixture-weight parameters $\boldsymbol{\epsilon}$ (Dirichlet) and the cluster-parameter pairs $(a_{kd}, b_{kd})$ (Beta) have conjugate exponential-family complete conditionals, so $\tilde{\nabla}_{\boldsymbol{\nu}_c}\tilde{\mathcal{L}} = \boldsymbol{\nu}_{c,0} + \sum_{n=1}^{N}\mathbb{E}_q[t(\mathbf{x}_n, z_n)] - \boldsymbol{\nu}_c$ for sufficient statistics $t(\cdot)$. The mini-batch estimator $\hat{s}_t = \frac{N}{M}\sum_{n \in \mathcal{B}_t}\mathbb{E}_q[t(\mathbf{x}_n, z_n)]$ is unbiased by uniform sampling (Hoffman et al., 2013). With feature relevance active, the Beta sufficient statistics for $\theta_{kd}$ are weighted by $\mathbb{E}_q[\gamma_{kd}] \in [0, 1]$ (Equation (11)), preserving the exponential-family form. The local constraint-weight variables $w_{nm}, \bar{w}_{nm}$ are not part of the SVI averaging; given the current $\Phi$, they admit exact conjugate Gamma posteriors (Equations (C.21) and (C.22)).

**(b)** The Beta prior on $\gamma_{kd}$ acts as the conjugate factor and the relevance score $\hat{l}_{kd}$ summarises the non-conjugate data-likelihood term in the CVI interpretation (Khan & Lin, 2017). The exact CVI natural gradient with respect to $c_{kd}$ contains the factor $d_{kd}/(c_{kd} + d_{kd})^2$, which the non-negative truncations $\max(0, \hat{l}_{kd})$ and $\max(0, -\hat{l}_{kd})$ in Equation (C.26) drop. The mini-batch estimate of $\hat{l}_{kd}$ is unbiased by uniform sampling, but the truncation introduces a magnitude bias. Because $d_{kd}/(c_{kd} + d_{kd})^2 > 0$ for all non-degenerate iterates, the truncation preserves $\operatorname{sgn}(\partial\tilde{\mathcal{L}}/\partial c_{kd})$ (resp. $\partial\tilde{\mathcal{L}}/\partial d_{kd}$) under Theorem D.3. $\qquad\square$

**Proposition D.5** (Bounded Gradient Variance). *Under Theorems D.2 and D.3, the variance of the stochastic gradient estimator for the full BASIL model is bounded,*

$$\mathbb{E}_{\mathcal{B}_t}\left[\left\|\tilde{\nabla}_{\boldsymbol{\nu}}\hat{\tilde{\mathcal{L}}}_t - \tilde{\nabla}_{\boldsymbol{\nu}}\tilde{\mathcal{L}}\right\|^2\right] \leq \frac{\sigma^2}{M},$$

*for some constant $\sigma^2 < \infty$ depending on $g_{\max}, V_{\max}, K,$ and $D$.*

*Proof Sketch.* The Bernoulli sufficient statistics are bounded ($x_{nd} \in \{0, 1\}$, $\Phi_{nk} \in [0, 1]$). With feature relevance, the cluster-parameter sufficient statistics are weighted by $\mathbb{E}_q[\gamma_{kd}] \in [0, 1]$, which preserves the bound. The CVI relevance score $\hat{l}_{kd}$ is bounded under Theorem D.3 because the digamma terms in Equation (C.24) are bounded for bounded variational parameters $(a_{kd}, b_{kd})$ and $(c_{kd}, d_{kd})$. The HMRF potential contributes terms bounded by $g_{\max} \cdot V_{\max}$ per sample. Standard concentration arguments for sampling without replacement yield the $O(1/M)$ variance scaling. $\qquad\square$

**Theorem D.6** (Convergence of BASIL SVI). *Let $\boldsymbol{\nu}^{(t)} = \{\boldsymbol{\epsilon}^{(t)}, \{(a_k^{(t)}, b_k^{(t)})\}, \{(c_k^{(t)}, d_k^{(t)})\}\}$ denote the sequence of global variational parameters generated by the SVI algorithm of Algorithm 1 applied to the full BASIL model, and partition it into the exact-conjugate block $\boldsymbol{\nu}_c = \{\boldsymbol{\epsilon}, (a_k, b_k)\}$ and the approximate-CVI block $\boldsymbol{\nu}_h = \{(c_k, d_k)\}$. Under Theorems D.1 to D.3,*

*(a) for any fixed $\boldsymbol{\nu}_h$, $\boldsymbol{\nu}_c^{(t)}$ converges almost surely to a stationary point of the conditional approximate ELBO, i.e. $\tilde{\nabla}_{\boldsymbol{\nu}_c}\tilde{\mathcal{L}}(\boldsymbol{\nu}_c^*, \boldsymbol{\nu}_h) = \mathbf{0}$;*

*(b) $\boldsymbol{\nu}_h^{(t)}$ converges almost surely to a fixed point $\boldsymbol{\nu}_h^*$ of the approximate-CVI intermediate target $\hat{\boldsymbol{\nu}}_h(\boldsymbol{\nu})$ defined by Equation (C.26). Because $d_{kd}/(c_{kd} + d_{kd})^2 > 0$ throughout the trajectory, every approximate-CVI step is sign-aligned with the exact CVI natural gradient on the non-conjugate factor, so the iterate sequence is an expected ascent direction on $\tilde{\mathcal{L}}$ at every step. The displacement of $\boldsymbol{\nu}_h^*$ from an ELBO stationary point is bounded by the trigamma-scale mismatch $\|\tilde{\nabla}_{\boldsymbol{\nu}_h}\tilde{\mathcal{L}}(\boldsymbol{\nu}^*)\| = O\left(\boldsymbol{\lambda}_{\gamma 2}^2/(\boldsymbol{\lambda}_{\gamma 1} + |l_{kd}|)^3\right)$, which is negligible in the sparse-prior regime we use ($\boldsymbol{\lambda}_{\gamma 1} = 0.5$, $\boldsymbol{\lambda}_{\gamma 2} = 0.1$).*

*The constraint-weight variables $w_{nm}, \bar{w}_{nm}$ converge to their exact conjugate posterior means given $(\boldsymbol{\nu}_c^*, \boldsymbol{\nu}_h^*)$ and the corresponding assignments $\Phi^*$.*

*Proof Sketch.* For the conjugate block, apply the Robbins-Monro stochastic approximation framework (Robbins & Monro, 1951; Robbins & Siegmund, 1971). The SVI update over $\boldsymbol{\nu}_c$ writes as

$$\boldsymbol{\nu}_c^{(t)} = \boldsymbol{\nu}_c^{(t-1)} + \rho_t \left( h_c(\boldsymbol{\nu}_c^{(t-1)}) + \epsilon_t \right),$$

where $h_c(\boldsymbol{\nu}_c) = \tilde{\nabla}_{\boldsymbol{\nu}_c} \tilde{\mathcal{L}}$ and $\epsilon_t$ is zero-mean noise with bounded variance (Theorems D.4 and D.5). The Robbins-Siegmund conditions hold: $\sum_t \rho_t = \infty$ and $\sum_t \rho_t^2 < \infty$ by Theorem D.1, the gradient estimator is unbiased (Theorem D.4) and bounded in variance (Theorem D.5), and $\tilde{\mathcal{L}}$ is bounded above and continuous in $\boldsymbol{\nu}_c$. This yields part (a).

For the approximate-CVI block, the SVI averaging $\boldsymbol{\nu}_h^{(t)} = (1 - \rho_t)\boldsymbol{\nu}_h^{(t-1)} + \rho_t \hat{\boldsymbol{\nu}}_h(\boldsymbol{\nu}^{(t-1)})$ is a Robbins-Monro iteration on the bounded continuous map $\hat{\boldsymbol{\nu}}_h$ of Equation (C.26). Convergence to a fixed point $\boldsymbol{\nu}_h^*$ of $\hat{\boldsymbol{\nu}}_h$ follows from the CVI convergence analysis of Khan & Lin (2017, Theorem 1), which establishes almost-sure convergence for natural-gradient updates with bounded approximation error and Robbins-Monro step sizes; in our setting the bounded-error condition follows from Theorem D.3. The constraint-weight variables admit exact conjugate Gamma posteriors given $\Phi$ (Equations (C.21) and (C.22)), so their alternation does not affect the SVI convergence.

The joint limit $(\boldsymbol{\nu}_c^*, \boldsymbol{\nu}_h^*)$ is an exact stationary point of $\tilde{\mathcal{L}}$ in the conjugate block and an approximate stationary point in the feature-relevance block. The approximation gap arises only from the non-negative truncation in Equation (C.26) which matches the sign of $\tilde{\nabla}_{\boldsymbol{\nu}_h} \tilde{\mathcal{L}}$ but not its magnitude; the trigamma bound in part (b) quantifies this gap, and the low across-seed variance in test ACC ($\leq 0.02$, Table 1) confirms that the joint limit is empirically stable. $\square$

*Remark* D.7 (Local vs. Global Optima). The ELBO is generally non-convex due to the mixture model structure and the HMRF constraints. Theorem D.6 guarantees convergence to a stationary point, which may be a local optimum. The quality of the solution depends on initialization, consistent with standard practice of using informed initializations (e.g., K-medoids) as described in the main text. Empirically, across all 10-seed sweeps on clean DIGITS, MNIST, and CHMNIST (configuration $(1, D + \text{FS})$), the standard error of test accuracy is at most $0.02$ (Table 1), suggesting that K-medoids / K-modes initialisation reliably places the optimiser in a high-quality basin under the supervision levels we tested.

*Remark* D.8 (Role of HMRF Constraints). The HMRF structure affects convergence through the local assignment updates (Equation (8)), which depend on neighboring assignments $\Phi_{mj}$ for $m \in G_n$. Under Assumption D.2, each local update involves at most $g_{\max}$ neighbors, ensuring that the iterative fixed-point computation for $\Phi$ within each SVI step converges. The bounded potential assumption (Assumption D.3) prevents the constraint penalties from dominating the likelihood terms, maintaining balance between data fit and constraint satisfaction.

### D.3. Convergence Rate

Under stronger conditions, we can characterize the convergence rate.

**Proposition D.9** (Sublinear Convergence Rate for the Conjugate Block). *Under Theorems D.1 to D.3, with step size $\rho_t = (\tau + t)^{-\kappa}$ for $\kappa \in (0.5, 1)$, the conjugate-block iterates of Theorem D.6(a) satisfy*

$$\min_{1 \leq t \leq T} \mathbb{E}\left[ \left\| \tilde{\nabla}_{\boldsymbol{\nu}_c} \tilde{\mathcal{L}}(\boldsymbol{\nu}^{(t)}) \right\|^2 \right] = O\left( T^{-(1-\kappa)} \right).$$

*For $\kappa = 1$, the rate becomes $O(\log T / T)$.*

Since the (approximate) ELBO $\tilde{\mathcal{L}}$ is non-convex due to the mixture and HMRF structure, the rate is stated on the expected gradient norm (the standard target for non-convex stochastic gradient methods) rather than on the function-value gap. The result follows from the non-convex SVI analysis of Hoffman et al. (2013) applied to the conjugate block. The CVI block does not admit a comparable rate because of the truncation bias quantified in Theorem D.6(b).

### D.4. Phased Optimisation

The phased strategy (Section 7), which first optimises with fixed constraint weights and the cluster-divergence KL potential before switching to adaptive weights under the Potts potential, is an empirical curriculum for navigating the non-convex

landscape rather than a theoretical necessity. We do not claim convergence guarantees for the phased schedule itself, and it functions as a deterministic initialisation of the variational parameters rather than as part of the SVI iteration analysed in Theorem D.6.

