# OpenReview forum: "BASIL: Scalable Bayesian Semi-supervised Clustering with Feature Selection and Adaptive Constraint Weighting"
_ICML.cc/2026/Conference — ICML 2026 regular_

### Official Review · Reviewer_ihTE · 2026-03-10

**Soundness:** 3
**Presentation:** 3
**Significance:** 2
**Originality:** 2
**Overall Recommendation:** 4
**Confidence:** 4

**Summary:**

This paper introduces BASIL, a scalable Bayesian semi-supervised clustering framework that utilizes stochastic variational inference to jointly infer cluster assignments and explicit feature importance. By incorporating an adaptive constraint-weighting mechanism to handle noisy supervision, the method demonstrates improved scalability, interpretability, and robust performance across synthetic, real-world, and large-scale health datasets.

**Compliance With Llm Reviewing Policy:**

Affirmed.

**Final Justification:**

The authors have addressed my concerns, and I decided to raise my scores.

**Key Questions For Authors:**

See the above weaknesses.

**Limitations:**

Yes

**Strengths And Weaknesses:**

Strengths:
1. By explicitly modeling feature relevance alongside cluster assignments, BASIL successfully addresses the black box nature of many existing scalable clustering algorithms, allowing users to understand which features drive the partitioning.
2. The introduction of an adaptive constraint-weighting mechanism is a highly practical contribution, effectively mitigating the negative impact of unreliable or contradictory prior knowledge common in real-world data.
3. Leveraging stochastic variational inference elegantly overcomes the traditional computational bottlenecks associated with Bayesian constrained clustering, enabling its use on large-scale datasets.

Weaknesses:
1. How does BASIL's joint inference perform when scaling to extremely high-dimensional feature spaces, such as raw high-resolution medical images? Does the explicit feature relevance modeling become a computational bottleneck in these extreme cases?
2. What are the main differences between the proposed method and existing methods? The current objective function appears more like a patchwork of existing approaches.
3. How exactly does the adaptive weighting mechanism distinguish between genuinely noisy constraints and simply complex, hard-to-cluster data points? Could this mechanism inadvertently discard valid but challenging prior knowledge?
4. Given the nature of variational inference in clustering tasks, how sensitive is the joint inference process to initial parameters or the initial subset of constraints? Are there empirical strategies provided to ensure the model avoids poor local optima?

---

> ### Author Rebuttal · Authors · 2026-03-30
>
> > **Q1: Computational scaling to high-dimensional features**
>
> The per-iteration cost is $O(MKD) + O(M|G|K)$, where $M$ is batch size, $K$ clusters, $D$ features, and $|G|$ average constraint neighbors. Which term dominates depends on the regime:
>
> When $|G| \gg D$ (dense supervision): constraints dominate. On DIGITS ($D$=64) at 50% supervision ($|G|$=358), training takes 44.4min vs 5.8s unsupervised, a 460$\times$ increase purely from constraints.
>
> When $D \gg |G|$ (high-dim, sparse supervision): feature dimension dominates. For raw 224$\times$224 images ($D$=50,176), the $O(MKD)$ term is ~784$\times$ larger than DIGITS. Feature importance (Eq. 4) has the same $O(MKD)$ cost as the base mixture update (Eq. 5), roughly doubling the matmul cost but not changing the scaling order.
>
> Practical strategies preserving pixel-level interpretability: (1) Resolution reduction (e.g., 56$\times$56, $D$=3,136) is standard in medical imaging (Yang et al., 2023). Our 28$\times$28 experiments (ChestMNIST, MNIST) demonstrate this. (2) The $O(MKD)$ operations ($\Phi^T X$) are GPU-parallelisable; porting from single-core NumPy to JAX/PyTorch gives estimated 10-50$\times$ speedup. Combined with moderate downsampling, high-resolution images become feasible within hours. These are engineering improvements that do not change the algorithm.
>
> > **Q2: Novelty and differences from existing methods**
>
> Our contributions are: (1) HMRF with learnable Gamma constraint weights (Eq. 3), whereas all prior methods (Basu et al., 2006; Echraibi et al., 2019) use fixed weights; (2) Beta-distributed feature selection (Eq. 4), novel in constrained clustering (confirmed by González-Almagro et al., 2025 survey); (3) Non-trivial SVI update rules (Propositions 5.3-5.4) derived for this specific model; (4) Ablation (Table B.8) reveals components interact non-trivially (e.g., $w_{nm}$ + KL instability, Appendix A.4), leading to the phased strategy (Section 6.1). This interaction analysis is itself a methodological contribution.
>
> > **Q3: Distinguishing noisy from hard-to-cluster constraints**
>
> This is a valid point. The mechanism is indirect: it detects constraint *violation*, not noise per se. The Gamma posterior down-weights constraints that persistently conflict with the learned clusters, regardless of whether they are noisy or simply hard. We will discuss this distinction explicitly in the revised paper.
>
> However, empirical evidence shows this indirect mechanism works because noise and violation are strongly correlated. Rescaled $\mathbb{E}[w_{nm}]$ over 10 runs, stratified by difficulty (top/bottom 25% feature distance) and ground truth:
>
> **DIGITS 20% noise**:
>
> | Hard clean | Hard corrupt | Easy clean | Easy corrupt | AUROC |
> |---|---|---|---|---|
> | 0.987 | 0.785 | 1.000 | 0.974 | 0.963 |
>
> **DIGITS 30% noise**:
>
> | Hard clean | Hard corrupt | Easy clean | Easy corrupt | AUROC |
> |---|---|---|---|---|
> | 0.909 | 0.791 | 0.989 | 0.989 | 0.808 |
>
> Within the same difficulty class, clean constraints consistently receive higher weight than corrupt ones (hard: 12-19 point gap). The blind spot is "easy corrupt" constraints (noisy but data-consistent), which retain high weight (0.974-0.989).
>
> We acknowledge that clustering in raw feature space limits this: the mean-field Bernoulli assumes feature independence (Appendix A.4), so some valid constraints requiring nonlinear separation may remain violated. This is the core trade-off: raw-feature interpretability (Figure 3) over representation flexibility.
>
> > **Q4: Sensitivity to initialisation and local optima**
>
> We agree this is an important practical consideration. We addressed it through: (1) Informed initialisation via K-medoids ($N$<1,000) or K-modes (Appendix A.2), providing data-driven starting points; (2) All results aggregated over 10 runs with different seeds (mean $\pm$ SE in all tables); (3) EMA convergence criterion (Eq. A.6) invariant to sample size; (4) Standard SVI schedule $\rho_t = (\tau+t)^{-\kappa}$ with $\kappa$=0.7, with convergence guarantees (Theorem D.3). The consistently low standard errors across experiments (e.g., ACC=0.87$\pm$0.00 on DIGITS at 50%, Table 1) confirm the model is not highly sensitive to initialisation in practice.

---

> > ### Author Rebuttal · Reviewer_ihTE · 2026-04-01
> >
> > Thanks to the authors' responses. The authors should illustrate why the joint integration of these distinct modules is fundamentally necessary to solve this specific clustering problem; the framework still appears as a combination of techniques. Therefore, I maintain the original evaluation.

---

> > > ### Author Response · Authors · 2026-04-02
> > >
> > > We thank the reviewer for this clarification. We agree that the ingredients of BASIL are connected to prior work. However, **our contribution is not a loose aggregation of modules, but a coupled Bayesian formulation for a specific problem setting**: semi-supervised clustering that must be **simultaneously scalable, feature-interpretable, and robust to noisy pairwise supervision**.
> > >
> > > **(1) The integration is model-level, not a patchwork of independent objectives.**
> > >
> > > BASIL optimizes a **single ELBO** from the joint model $p(X,\mathbf{z},\Theta\mid Y)$, where $\Theta=(\pi,\theta,\gamma,W,\bar W)$. The feature relevance variables $\gamma_{kd}$ enter the likelihood through $\tilde{\theta}\_{kd}=\gamma\_{kd}\theta_{kd}+(1-\gamma_{kd})\theta_{0d}$, so removing $\gamma$ changes the **data-generating process**, not merely a regularizer. Likewise, $w_{nm},\bar w_{nm}$ are **latent variables with Gamma posteriors**, not fixed penalties. The updates are **circularly coupled**: $\gamma$ changes $\theta$; $\theta$ determines HMRF cluster divergences; these divergences determine the posteriors of $w_{nm},\bar w_{nm}$; the learned weights enter the assignment update $\Phi_{nk}$; and the updated $\Phi$ drives feature relevance again. This is **joint posterior inference**, not modular stacking.
> > >
> > > **(2) A separated treatment would remove exactly the information needed here.**
> > >
> > > If feature relevance were learned separately, it would no longer be tied to the final **constraint-consistent partition**. If constraint reliability were estimated separately, it would not see the evolving **cluster geometry**. If both were fixed before scalable inference, the model could not revise them jointly using unlabeled data. This is why we view the unified treatment as important, consistent with our discussion that decoupling representation/metric learning from clustering fails to fully exploit global unlabeled structure.
> > >
> > > **(3) The empirical behavior is interaction-dependent, not what one would expect from loosely separable add-ons.**
> > >
> > > DIGITS ablation at **20% supervision**:
> > >
> > > | Config | Clean | 20% noise | 50% noise |
> > > |---|---:|---:|---:|
> > > | `(1,--)` | 0.80±0.00 | 0.76±0.02 | 0.68±0.01 |
> > > | `(1,FS)` | 0.80±0.02 | 0.76±0.04 | 0.69±0.00 |
> > > | `(w,--)` | 0.76±0.02 | 0.75±0.02 | 0.68±0.01 |
> > > | `(w,FS)` | 0.78±0.04 | **0.80±0.03** | **0.74±0.05** |
> > >
> > > Under **clean constraints**, fixed-weight variants are already competitive. Under **noisy constraints**, adaptive weighting helps **only when paired with feature selection**: at **20% noise**, `(w,FS)` reaches **0.80±0.03**, outperforming the other three. At **50% noise**, `(w,FS)` reaches **0.74±0.05**, while the other three remain at `0.68-0.69`. If the components were independent add-ons, their effects should be more nearly **additive and stable**; instead, their benefit is clearly **interaction-dependent**.
> > >
> > > **(4) Additional rebuttal analysis supports this interpretation.**
> > >
> > > In Q3 we examined whether adaptive weighting simply discards difficult constraints. We observe that this is **not** the case: within the same difficulty class, **hard clean constraints still receive higher weights than hard corrupted constraints**, so the mechanism is not merely penalizing “hardness.” In an additional analysis conducted for the rebuttal, we also observe that adaptive weighting **alone** only weakly identifies corrupted constraints, whereas adaptive weighting **with feature selection** yields much stronger discrimination:
> > >
> > > | Config | 20% noise AUROC | 50% noise AUROC |
> > > |---|---:|---:|
> > > | `(1,--)` | 0.500 | 0.500 |
> > > | `(1,FS)` | 0.500 | 0.500 |
> > > | `(w,--)` | 0.630 | 0.802 |
> > > | `(w,FS)` | **0.953** | **0.914** |
> > >
> > > This is consistent with the interpretation above: feature selection sharpens cluster geometry, making violations more informative for reliability estimation; the resulting weights then help protect clustering under heavier supervision noise.
> > >
> > > **(5) The target applications also motivate this joint treatment.**
> > >
> > > Our healthcare use cases combine **imperfect supervision** with a need for **feature-level interpretation**. In this regime, “cluster first, explain later” or weighting constraints independently is inadequate. BASIL therefore **jointly learns assignments, feature relevance, and constraint reliability**, so the resulting clusters are both interpretable and robust.
> > >
> > > Finally, our ablations also show that naive combinations are **not** automatically stable: KL-based potentials and adaptive weight learning can interact poorly, which motivated our modified potential and phased optimization strategy. This further supports viewing BASIL as a **nontrivial coupled inference framework**, rather than a simple combination of techniques.
> > >
> > > We will revise the paper to make this distinction clearer: BASIL does not claim that every ingredient is new in isolation, but that a **unified Bayesian treatment is important to jointly achieve scalability, interpretability, and robustness** in the constrained clustering setting we study.

---

### Official Review · Reviewer_qT4M · 2026-03-11

**Soundness:** 3
**Presentation:** 3
**Significance:** 2
**Originality:** 2
**Overall Recommendation:** 5
**Confidence:** 4

**Summary:**

The paper proposes BASIL, a Bayesian semi-supervised clustering framework that incorporates must-link and cannot-link constraints through an HMRF prior, performs scalable inference with stochastic variational inference, learns cluster-specific feature importance weights, and adaptively down-weights unreliable constraints via Gamma-distributed latent weights.

**Compliance With Llm Reviewing Policy:**

Affirmed.

**Final Justification:**

My concerns have been addressed in the rebuttal, thus I increased my score.

**Key Questions For Authors:**

1) The current model is limited to binary features. How difficult would it be to extend BASIL to continuous or mixed data types?
2) Why were recent deep constrained clustering methods not included in the empirical comparisons? How does BASIL perform relative to modern deep generative clustering approaches?
3) The paper presents BASIL as a Bayesian method with adaptive latent constraint weights, but the experimental section does not directly assess the quality of the resulting uncertainty estimates. Additional analysis on this aspect would help substantiate the Bayesian formulation.
4) The interpretability claim is central to the paper, but most of the supporting evidence currently comes from qualitative heatmaps and clinical narratives. The paper would be strengthened by adding a quantitative evaluation of feature relevance quality.

**Limitations:**

Yes.

**Strengths And Weaknesses:**

**Strength**

1)	Soundness: Overall, the modeling framework is technically reasonable and builds upon established Bayesian clustering literature. It defines a Bayesian generative model with mixture proportions, component parameters, feature-selection variables, and learnable must-link and cannot-link weights. The Inference of this model is performed with SVI using minibatches.

2)	Presentation: The presentation of this manuscript is generally clear and well-organized.

3)	Significance: The method is aimed at a practical but relatively underexplored setting: semi-supervised clustering with pairwise constraints, where it is also desirable to retain some interpretability over feature importance and to scale beyond full-batch Bayesian inference.

4)	Originality: Although the paper is not introducing a wholly new paradigm, the combination of Bayesian constrained clustering, SVI, feature selection and adaptive reliability weighting is an interesting and meaningful idea.

**Weakness**

1)	The main benchmark section compares against MML, PCK, and MPCK, while the related-work section itself discusses more modern deep/Bayesian constrained clustering alternatives, but the empirical section does not compare against them.

2)	The manuscript argues that Bayesian modeling is attractive for uncertainty, but the reported results focus mostly on ACC, runtime, or qualitative feature maps. It would be helpful to include posterior uncertainty directly in evaluation, especially since adaptive constraint weighting is a central claim.

3)	The paper emphasizes interpretability through feature heatmaps, but the current evidence is primarily qualitative. A more rigorous quantitative evaluation of feature relevance would further support the interpretability claims.

---

> ### Author Rebuttal · Authors · 2026-03-30
>
> > **Q1: Broader empirical comparison**
>
> Please see our response to Reviewer tmhs (Q1). We will add DCGMM (Manduchi et al., 2021) as a deep baseline with quantitative comparison tables.
>
> > **Q2: Posterior uncertainty evaluation**
>
> We provide new quantitative validation of BASIL's uncertainty estimates of constraint weights. Using DIGITS with controlled noise (10-50% label corruption), we compare rescaled learned $\mathbb{E}[w_{nm}]$ between clean and corrupted constraints:
>
> | Noise | Test ACC | $\mathbb{E}[w]$ clean | $\mathbb{E}[w]$ corrupt | AUROC |
> |---|---|---|---|---|
> | 0% | 0.80$\pm$0.03 | 0.999 | n/a | n/a |
> | 10% | 0.77$\pm$0.04 | 0.997 | 0.914 | 0.969$\pm$0.012 |
> | 20% | 0.76$\pm$0.02 | 0.997 | 0.872 | 0.953$\pm$0.017 |
> | 30% | 0.76$\pm$0.03 | 0.996 | 0.852 | 0.947$\pm$0.010 |
> | 50% | 0.71$\pm$0.04 | 0.992 | 0.858 | 0.914$\pm$0.016 |
>
> Clean constraints retain ~100% weight while corrupted ones are progressively down-weighted (91% to 86%). AUROC degrades gracefully from 0.97 to 0.91. This is unique to BASIL: no baseline provides per-constraint reliability estimates.  $\mathbb{E}[w]$ is the direct Bayesian output that modulates each constraint's influence during inference, unlike non-Bayesian baselines (PCK, MPCK, DC-GMM), which use fixed uniform weights with no mechanism to express per-constraint reliability. The rescaling to [0,1] via the prior mean provides an interpretable scale where 1.0 = full prior confidence and lower values = data-driven downweighting.
>
> > **Q3: Quantitative feature relevance**
>
> We provide quantitative evaluation using synthetic data (Table B.5) with known feature roles. We extract $\max_k \mathbb{E}[\gamma_{kd}]$ from trained models (Table B.6) across 10 runs:
>
> | Setting | Informative | Irrelevant | Noninformative | AUROC |
> |---|---|---|---|---|
> | N=500, 20% | 0.971 | 0.171 | 0.043 | 1.00 |
> | N=5k, 20% | 0.975 | 0.005 | 0.005 | 1.00 |
> | N=50k, 20% | 0.997 | 0.663 | 0.020 | 1.00 |
>
> Perfect AUROC (1.00) for distinguishing informative from noninformative features across all sizes. Irrelevant features receive intermediate $\gamma$ at $N$=50k (0.66), which is correct: they vary across components (Table B.5), so $\gamma_{kd}$ correctly identifies them as cluster-discriminative (Section 4).
>
> > **Q4: Extension to continuous/mixed data**
>
> Please see our response to Reviewer YaKw (Q1) for the concrete 5-step Gaussian extension. For mixed types: product likelihoods (Bernoulli $\times$ Gaussian) with independent conjugate priors per feature.

---

> > ### Author Rebuttal · Reviewer_qT4M · 2026-04-04
> >
> > The authors have addressed my previous concerns, and I will raise my score accordingly.

---

### Official Review · Reviewer_tmhs · 2026-03-11

**Soundness:** 3
**Presentation:** 4
**Significance:** 3
**Originality:** 3
**Overall Recommendation:** 5
**Confidence:** 4

**Summary:**

This paper proposes a Bayesian semi-supervised clustering method that solves scalability, interpretability, and robustness to noisy supervision issues by combining the HMRF formulation with stochastic variational inference, feature relevance learning, and adaptive constraint weighting.

**Compliance With Llm Reviewing Policy:**

Affirmed.

**Final Justification:**

The authors have addressed my concerns well.

**Key Questions For Authors:**

(1) Replace the comparison method with a newer one to highlight the method's advantages.
(2) Formulas are not labeled; the overall format needs to be checked.
(3) There are slightly fewer recent publications; some relevant citations need to be added.

**Limitations:**

yes

**Strengths And Weaknesses:**

The paper demonstrates a clear line of thought, fluent language, and considerable innovation. The problems are that the formulas are not numbered, there are few recent publications, and the comparison methods are not very new.

---

> ### Author Rebuttal · Authors · 2026-03-30
>
> > **Q1: Newer comparison methods**
>
> We have compared BASIL with DC-GMM (Manduchi et al., NeurIPS 2021) as a deep generative baseline.  We use their open-source code with a reduced FC architecture (128-64-32-10) suited to DIGITS' 64-dim input, pretrained from scratch, and feed our exact constraint matrices. All models use the same 10 random seeds, CPU-only for fair comparison.
>
> **Table 1 (clustering accuracy and time):**
>
> | Model | 0% sup | 20% sup | 50% sup |
> |---|---|---|---|
> | BASIL ACC | 0.69$\pm$0.02 | 0.80$\pm$0.02 | 0.87$\pm$0.00 |
> | BASIL Time | 5.81s$\pm$0.34 | 7.00m$\pm$0.25 | 44.4m$\pm$1.20 |
> | DC-GMM ACC | 0.16$\pm$0.02 | 0.51$\pm$0.30 | 0.61$\pm$0.38 |
> | DC-GMM Time | 72s$\pm$2 | 136s$\pm$3 | 140s$\pm$3 |
>
> **Figure 4 (test ACC, noise robustness):**
>
> | Noise | BASIL 20% | DC-GMM 20% | BASIL 50% | DC-GMM 50% |
> |---|---|---|---|---|
> | 0% | 0.80$\pm$0.02 | 0.52$\pm$0.32 | 0.87$\pm$0.00 | 0.61$\pm$0.38 |
> | ~10% | 0.83$\pm$0.00 | 0.49$\pm$0.34 | 0.84$\pm$0.00 | 0.47$\pm$0.34 |
> | ~20% | 0.80$\pm$0.03 | 0.74$\pm$0.22 | 0.82$\pm$0.00 | 0.22$\pm$0.22 |
> | ~30% | 0.81$\pm$0.02 | 0.33$\pm$0.30 | 0.84$\pm$0.00 | 0.29$\pm$0.27 |
> | ~50% | 0.74$\pm$0.05 | 0.21$\pm$0.18 | 0.72$\pm$0.05 | 0.22$\pm$0.20 |
>
> DC-GMM is faster per run (~2min vs 7-44min) because it processes constraints via vectorized sparse tensor operations (TF/C++ backend), while BASIL's current implementation uses a Python loop over per-sample constraint neighbors (Eq. 2). This is an implementation bottleneck, not algorithmic. Despite the speed advantage, DC-GMM is highly unstable (SE 0.18-0.38 vs BASIL's 0.00-0.05), with 30-50% of runs collapsing. It degrades sharply under noise, especially at 50% supervision where more noisy constraints are available.  DC-GMM also lacks feature-level interpretability (cf. Figure 3), adaptive constraint weighting (cf. Eq. 3), and posterior uncertainty quantification.
>
> **MNIST (N=60,000, binary; DC-GMM: alpha=100 BCE, 10 seeds):**
>
> | Model | 0% sup | 20% sup | 50% sup |
> |---|---|---|---|
> | BASIL test ACC | 0.59$\pm$0.02 | 0.68$\pm$0.02 | 0.79$\pm$0.00 |
> | BASIL time | 18.7m$\pm$0.41 | 1.52h$\pm$0.02 | 1.57h$\pm$0.01 |
> | DC-GMM test ACC | 0.114$\pm$0.000 | 0.508$\pm$0.125 | 0.722$\pm$0.066 |
> | DC-GMM time | 3h$\pm$0.2 | 3h$\pm$0.1 | 3h$\pm$0.1 |
>
> Key findings on MNIST: DC-GMM requires substantial hyperparameter tuning (alpha=100 vs. default alpha=10000) and even then fails 50% of seeds at 20% supervision due to sensitivity to GMM initialization in the latent space.  At 50% supervision, DC-GMM's ACC is 0.789 vs. BASIL's 0.79, which is comparable but at ~2x the runtime and with much higher variance.
>
> > **Q2: Equation labeling**
>
> We will number all equations in the revised manuscript. Key equations (1)-(5) are numbered; intermediate displays will be converted for easier referencing.
>
> > **Q3: Recent publications**
>
> Many citations are necessarily older because Bayesian constrained clustering builds on foundational work from the 2000s (COP-Kmeans, PCK-means, HMRF, SVI), and the field has seen relatively few Bayesian contributions since. We already cite the comprehensive 2025 survey (Gonzalez-Almagro et al.) and recent relevant deep methods (Manduchi 2021, Hazratgholizadeh 2022). We will add more recent related work: Rao & Kirk (2025, *Bioinformatics Advances*) on variational Bayesian clustering with variable selection, and Riverain et al. (2022, *Machine Learning*) on probabilistic semi-supervised clustering with pairwise constraints via HMRF and variational EM.

---

> > ### Author Rebuttal · Reviewer_tmhs · 2026-04-02
> >
> > The authors have addressed my concerns. Thus, I will raise my scores.

---

### Official Review · Reviewer_YaKw · 2026-03-15

**Soundness:** 3
**Presentation:** 3
**Significance:** 3
**Originality:** 3
**Overall Recommendation:** 4
**Confidence:** 3

**Summary:**

The paper presents BASIL, a Bayesian semi-supervised clustering framework built on an HMRF formulation with must-link and cannot-link constraints, stochastic variational inference, cluster-specific feature relevance learning, and adaptive latent weights for pairwise constraints. The main goals are to improve scalability relative to prior Bayesian constrained clustering methods, provide interpretable feature-level explanations for clusters, and improve robustness to noisy or inconsistent supervision. The empirical evaluation covers synthetic binary data, digit benchmarks, noisy-constraint settings, and large-scale healthcare applications. The paper also provides a partial theoretical treatment and is explicit about limitations of the full formulation. Overall, the submission’s broad aspect pertains to scalable and interpretable constrained clustering under uncertainty. Overall, this submission’s notable idea concerns the joint learning of cluster assignments, feature relevance, and adaptive constraint reliability in a single Bayesian framework.

**Compliance With Llm Reviewing Policy:**

Affirmed.

**Key Questions For Authors:**

1. Please separate more explicitly what is theoretically established from what is heuristic in the full model. My score would increase if this boundary were made sharper and if the heuristic updates were justified more clearly in practice.
2. Since the Bernoulli likelihood is a major restriction, can the paper provide at least one concrete extension path to mixed or continuous data, beyond noting it as future work?
3. Please provide a clearer component analysis under matched compute. How much of the gain comes from stochastic variational inference, how much from feature selection, and how much from adaptive constraint weighting?
4. The noisy-constraint results are interesting but nuanced. Please summarize more directly when adaptive constraint weighting helps, when it hurts, and how sensitive it is to hyperparameter settings.
5. A stronger comparison to modern deep constrained clustering baselines would improve the significance claim, even if those methods are less interpretable.

**Limitations:**

Yes

**Strengths And Weaknesses:**

- The problem is worthwhile. Combining semi-supervised clustering, uncertainty quantification, interpretability, and robustness to noisy constraints is important in applications such as healthcare.
- The paper makes a clear modeling contribution. It does not merely add stochastic variational inference for scalability, but also integrates feature relevance and adaptive constraint weighting into a Bayesian constrained-clustering setup.
- The healthcare case studies are a real strength. They make the paper feel application-driven rather than purely benchmark-driven, and they are aligned with the interpretability motivation.
- The paper is honest about limitations. It explicitly discusses the Bernoulli-data restriction, mean-field assumptions, heuristic aspects of the full optimization scheme, and instability when all components are jointly optimized.
- The interpretability angle is well motivated. The feature-importance analyses and cluster-profile discussions are more informative than what most constrained clustering baselines provide.

Weaknesses.
- The method is currently restricted to binary data. This is a substantial limitation for a paper making relatively broad claims about practical constrained clustering.
- The theoretical treatment does not fully cover the model emphasized in experiments. The full BASIL pipeline still relies on approximate updates and phased optimization, and the paper itself notes that a complete theory remains unresolved.
- Robustness to noisy supervision is promising but not uniform. The method appears helpful in some noisy regimes, but the combined model can also become less attractive when supervision is already relatively clean.
- The runtime story is mixed. The method scales better than some Bayesian baselines on larger problems, but simpler baselines can still be faster on smaller benchmarks.
- The comparison set could be broader, especially with respect to more recent deep constrained clustering methods, even if the paper intentionally prioritizes interpretability and uncertainty quantification.

The paper is substantial, careful, and application-motivated, though some of the strongest claims are limited by the binary-data restriction and by the gap between the full empirical pipeline and the current theoretical treatment. The high-level contributions are easy to understand, and the paper generally does a good job distinguishing what is and is not fully solved. A scalable Bayesian constrained clustering method with explicit interpretability could be useful, especially in health-related applications where transparency matters. The combination of stochastic variational inference, adaptive constraint weighting, and feature selection is interesting and reasonably novel.

---

> ### Author Rebuttal · Authors · 2026-03-30
>
> > **Q1: Binary data restriction & concrete extension path**
>
> Our model is generic and we have derived the concrete Gaussian extension:
> 1. Replace Bernoulli with Gaussian; Beta prior with Normal-Inverse-Gamma (standard conjugate pair; Bishop, 2006).
> 2. Feature shrinkage (Section 4) carries over identically.
> 3. Mixture parameter update remains closed-form conjugate.
> 4. Feature importance heuristic (Eq. 4) applies with relevance score evaluated at current parameters.
> 5. KL-divergence potential (Eq. 1) has closed form for Gaussians.
>
> Algorithm 1 and all other updates ($\pi, w_{nm}, \Phi_{nk}$) are unchanged. For mixed data: product likelihoods (Bernoulli x Gaussian) per feature.
>
> > **Q2: Theory vs. heuristic boundary**
>
> We will sharpen this boundary. **Theoretically grounded**: (i) SVI convergence (Appendix D, Theorem D.3: a.s. convergence via Robbins-Monro, Prop. D.4: $O(T^{-(1-\kappa)})$ rate); (ii) constraint weight updates (Eq. 3) are exact coordinate ascent for the Gamma rate; (iii) mixture parameter updates (Eq. 5) are standard conjugate; (iv) feature importance (Eq. 4) is a conjugate-computation VI step (Khan & Lin, 2017) for the non-conjugate $\gamma_{kd}$: $l_{kd}$ provides pseudo-sufficient-statistics within the SVI update $\nu^{(t)}=(1-\rho_t)\nu^{(t-1)}+\rho_t(\lambda_0+\frac{N}{M}\hat{s})$, extending Theorem D.3 to $(\theta\_k,\gamma_k,\pi,w\_{nm},\bar{w}\_{nm})$ under first-order expectation approximation. **Remaining heuristic**: (i) phased optimisation (Section 6.1), empirically calibrated.
>
> Minor typo in Appendix C gradient: $2(\psi(c+d)-\psi(c))$ terms should not appear, but do not affect Eq. 4 (only $\mathrm{sign}(l_{kd})$ matters).  We will also update the theoretical convergence analysis in Appendix D and extend it to the full model by considering CVI (Khan & Lin, 2017) connections.
>
> > **Q3: Robustness to noisy supervision**
>
> Adaptive weighting helps when data is small and noise is moderate-to-high; it does not help (and slightly hurts variance) when data is large with sparse constraints or when constraints are clean.
>
> **HELPS** (small data + noise): On DIGITS (N=1,437) with 20% noisy constraints at 20% supervision (Table B.10), tuned $(w_{nm}$,FS) $Gamma(1,2.5)$ achieves ACC=0.80 vs fixed (1,FS) ACC=0.76 (+0.04). At 30% noise (Table B.14), BASIL=0.81 vs PCK=0.77.
>
> **DOES NOT HELP** (large data, sparse constraints): On MNIST (N=60k) with 20% noise (Table B.17), adaptive ACC=0.65 vs fixed ACC=0.66. *Root cause*: constraints capped at 5M/72M pairs (7%). Each sample sees ~7 constraint neighbors/batch vs ~100 for DIGITS, too few for the Gamma posterior to concentrate. This is a practical limitation of the constraint budget, not fundamental to the algorithm.
>
> **HURTS** (clean): On DIGITS with clean constraints (Table B.8), adaptive shows higher SE. At low noise (<1%, Table B.9), fixed ACC=0.84 matches adaptive with lower variance.
>
> **Hyperparameter**: $\lambda_{w2}$ is the key knob. From Tables B.9-B.11 in Appendix: $\lambda_{w2}$=1 for clean; $Gamma(1,1.5)$ for <1% noise; $Gamma(1,2.5)$ for 20%+. Sensitivity is moderate: $\lambda_{w2}\in[1.5,5]$ covers most regimes.
>
> > **Q4: Broader comparison with deep methods**
>
> Please see our response to Reviewer tmhs (Q1). We add DCGMM (Manduchi et al., 2021) as a deep baseline with quantitative comparison tables.
>
> > **Q5: Runtime and scalability**
>
> The scalability claim is for large $N$. On MNIST ($N$=60k), all baselines exceed 2 days while BASIL completes in 1.52h (>96% reduction, Table 1). On synthetic $N$=50k, 19.42min vs >2 days for PCK/MPCK (Table B.7). On small DIGITS ($N$=1,437), PCK (9.92s) can be faster than BASIL (7.00min at 20%). SVI benefits when $N \gg M$ (Hoffman et al., 2013): each iteration costs $O(M)$ vs $O(N)$. For detailed complexity analysis, see our response to Reviewer ihTE (Q1).
>
> > **Q6: Component analysis under matched compute**
>
> SVI is the scalability enabler (slightly better accuracy in 2.5% of compute at N=50k); FS accelerates convergence without hurting accuracy; adaptive $w_{nm}$ provides accuracy gains under noise at matched compute.
>
> **SVI**: Tables B.3-B.4, synthetic N=50k/20%: BASIL NMI=0.78 in 13.84min vs full-batch NMI=0.73 in 9.10h. Full-batch cannot scale beyond N=50k; BASIL handles N=500k in 24min.
>
> **FS**: Table B.8, DIGITS 20%: (1,FS) ACC=0.80 in 7.04min vs (1,--) ACC=0.80 in 8.42min (16% less compute). At 50%: both ACC=0.87, 0.74h vs 0.89h (17% faster). FS's main gain is convergence speed and interpretability (Figure 3).
>
> **Adaptive $w_{nm}$**: Table B.10, DIGITS with 20% noisy constraints at 20% supervision: tuned $(w_{nm}$,FS) ACC=0.80 vs fixed (1,FS) ACC=0.76 (+0.04) in comparable compute (6.75min vs 7.30min).

---

### Decision · Program_Chairs · 2026-04-30

**Decision:**

Accept (regular)

**Comment:**

The paper introduces a Bayesian semi-supervised clustering framework. It involves an HMRF formulation with must-link and cannot-link constraints, stochastic variational inference, cluster-specific feature relevance learning, and adaptive latent weights for pairwise constraints. The review confirms the problem, design, contributions, and use cases, while also raises spaces for further clarify, expansion, evaluation, such as on data settings, robustness, and evaluation. The authors may consider to address the comments in the final version.